

# Large Regional Differences in Antarctic Ice Shelf Mass Loss from Southern Ocean Warming and Meltwater Feedbacks

Morven Muilwijk[1], Tore Hattermann[1], Rebecca L. Beadling[2], Neil C. Swart[3], Aleksi Nummelin[4,6], Chuncheng Guo[5,6], David M. Chandler[6], Petra Langebroek[6], Shenjie Zhou[7], Pierre Dutrieux[7], Jia-Jia Chen[8], Christopher Danek[9], Matthew H. England[10,11], Stephen M. Griffies[12], F. Alexander Haumann[9,13], André Jüling[14], Ombeline Jouet[9], Qian Li[15], Torge Martin[16], John Marshall[15,17], Andrew G. Pauling[18], Ariaan Purich[19], Zihan Song[20], Inga J. Smith[18], Max Thomas[21], Irene Trombini[22, 23], Eveline van der Linden[14], and Xiaoqi Xu[24, 25]

[1]Norwegian Polar Institute, Fram Centre, Tromsø, Norway
[2]Department of Earth and Environmental Science, Temple University, Philadelphia, PA, USA
[3]Canadian Centre for Climate Modelling and Analysis, Environment and Climate Change Canada, Victoria, BC, Canada
[4]Finnish Meteorological Institute, Helsinki, Finland
[5]Danish Meteorological Institute, Copenhagen, Denmark
[6]NORCE Research AS, Bjerknes Centre for Climate Research, Bergen, Norway
[7]British Antarctic Survey, Cambridge, UK
[8]School of Earth and Atmospheric Sciences, Georgia Institute of Technology, USA
[9]Alfred Wegener Institute, Helmholtz Centre for Polar and Marine Research, Bremerhaven, Germany
[10]Centre for Marine Science and Innovation (CMSI), University of New South Wales, Sydney, Australia
[11]ARC Centre for Excellence in Antarctic Science, University of New South Wales, Sydney, Australia
[12]Princeton University Atmospheric and Oceanic Sciences Program, NJ, USA
[13]Ludwig-Maximilians-Universität München, Munich, Germany
[14]Royal Netherlands Meteorological Institute, De Bilt, The Netherlands
[15]Department of Earth, Atmospheric, and Planetary Sciences, Massachusetts Institute of Technology, Cambridge, MA, USA
[16]GEOMAR Helmholtz Centre for Ocean Research Kiel, Kiel, Germany
[17]NASA Goddard Institute for Space Studies, New York, NY, USA
[18]Department of Physics, University of Otago, Dunedin, New Zealand
[19]School of Earth, Atmosphere and Environment, ARC Special Research Initiative for Securing Antarctica's Environmental Future, Monash University, Melbourne, Australia
[20]Earth System Physics Section, Abdus Salam International Centre for Theoretical Physics, Trieste, Italy
[21]Met Office Hadley Center, Exeter, UK
[22]Department of Physics and Astronomy, University of Bologna, Bologna, Italy
[23]Institute of Atmospheric Sciences and Climate, National Research Council of Italy, Bologna, Italy
[24]Key Laboratory of Earth System Numerical Modelling and Application, Institute of Atmospheric Physics, Chinese Academy of Sciences, Beijing, China
[25]University of Chinese Academy of Sciences, Beijing, China

**Correspondence:** Morven Muilwijk (morven.muilwijk@npolar.no)

**Abstract.** The increasing release of Antarctic meltwater represents one of the most profound, yet uncertain, consequences of global climate change. The absence of interactive ice sheets in state-of-the-art climate models prevents the direct calculation of ice-ocean feedbacks, leaving significant uncertainty in the global and regional consequences of meltwater discharge. This study leverages results from the Southern Ocean Freshwater Input from Antarctica (SOFIA) initiative to assess the ocean response to a 0.1 Sv meltwater perturbation and to infer the resulting feedback on ice shelf basal melting across 10 CMIP6 models. We analyze meltwater-induced temperature anomalies across distinct continental shelf regimes, compare them with SSP5-8.5 global warming-induced anomalies, and translate these into basal melt rates using a parameterization calibrated with a new observational climatology. Although the meltwater feedback is generally thought to amplify basal melting, our results demonstrate large regional differences, with implied enhanced ice shelf mass loss in some sectors but suppressed basal melting in others. The model ensemble indicates a warming feedback on the continental shelf in most East Antarctic regions, whereas in West Antarctica, the region with the greatest observed ice shelf mass loss in recent decades, most models simulate





cooling or reduced warming, suggesting a negative feedback. This regional contrast implies that East Antarctica may play an increasingly dominant role in future ice shelf mass loss. Simulations support existing hypotheses linking these asymmetric temperature responses to strong regional connectivity and shelf break dynamics, including a strengthened Antarctic Slope

Front, an accelerated Antarctic Slope Current, and reduced dense shelf water formation.

## 1   Introduction

The release of Antarctic meltwater represents one of the most profound yet uncertain consequences of future global climate change. Observational evidence reveals that the Antarctic Ice Sheet (AIS) and its ice shelves are undergoing significant mass loss (Adusumilli et al., 2020; Otosaka et al., 2023; Paolo et al., 2023; Davison et al., 2023, 2025), with the rate of loss

accelerating particularly in regions undergoing rapid ice shelf melting (Paolo et al., 2015; Rignot et al., 2019). Projections from standalone ice sheet models indicate that this mass loss will continue to accelerate in response to anthropogenic greenhouse gas forcing (Seroussi et al., 2020, 2024), leading to further increases in meltwater discharge into the Southern Ocean. This mass loss is expected to become the primary contributor to global sea level rise in the coming decades and centuries (Edwards et al., 2021; Fox-Kemper et al., 2021), while associated meltwater also significantly affects regional and global climate (Bronselaer

et al., 2018; Fyke et al., 2018; Golledge et al., 2019; Rye et al., 2020; Dong et al., 2022; Purich and England, 2023; Beadling et al., 2024; Fricker et al., 2025; Xu et al., 2025).

Because most coupled climate models, including those in the latest Coupled Model Intercomparison Project (CMIP6, Eyring et al., 2016), do not include fully interactive ice sheets and ice shelves, substantial uncertainty remains about the magnitude (Bamber et al., 2019; Levermann et al., 2020; Edwards et al., 2021) and impacts (Swart et al., 2023; Lambert et al., 2024) of

meltwater discharge. This limitation prevents explicit calculation of ice-sheet–ocean–atmosphere feedback mechanisms linked to meltwater discharge, which represents a major source of uncertainty in future climate projections (Fyke et al., 2018; Golledge et al., 2019; Sadai et al., 2020; Lambert et al., 2024).

This problem is not new; over the past few decades, numerous studies have explored this topic on various spatial and temporal scales. A comprehensive list of these studies is provided in Swart et al. (2023). These studies typically rely on freshwater

perturbation experiments, often referred to as "hosing experiments," where the ocean is forced with additional freshwater using coupled climate models or ocean–sea-ice-only simulations. A key limitation of these studies is the lack of consistency in the experimental design. Approaches vary widely in the magnitude, spatial and temporal distribution of freshwater forcing, and the methods used to impose freshwater and heat fluxes associated with ice melt. Most models do not include ice shelves and icebergs, representing meltwater as runoff from the continent and simulating its entry into the ocean at the surface rather than

at depth. Moreover, most Southern Ocean hosing experiments have been conducted using single models, ranging from simple theoretical frameworks and simplified intermediate-complexity models, to fully-coupled Earth System Models with varying but generally coarse spatial resolutions. As a result, the findings across these existing studies are highly model-dependent, often yielding divergent or even contradictory conclusions regarding responses on the continental shelf. Additionally, the diversity in experimental designs, including the background climate ranging from preindustrial to various global warming scenarios, and

freshwater forcing styles not only complicates direct comparisons between studies but also hinders the assessment of model spread, which is typically achieved through coordinated intermodel comparison projects, e.g., the CMIP framework (Eyring et al., 2016).

Despite variations across previous studies, certain responses to additional meltwater appear qualitatively robust. Offshore from the continental break, the Southern Ocean water column is characterized by a cold surface layer overlying a warmer deep

layer known as Circumpolar Deep Water (CDW). Models consistently show that additional freshwater reduces surface water density, strengthening water column stratification and isolating the (relatively) warm CDW from the surface where atmospheric





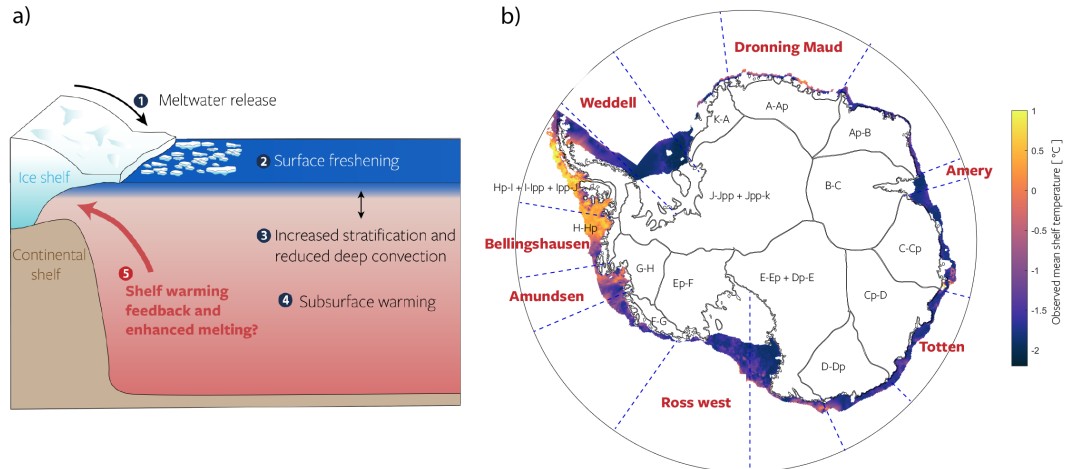

**Figure 1.** a) Schematic illustrating the impact of Antarctic Ice Sheet meltwater on Southern Ocean hydrography, emphasizing the potential subsurface ocean warming and ice shelf melting feedback mechanisms explored in this study. b) Antarctic Ocean sectors (blue dashed lines) used in this study, based on definitions by Jourdain et al. (2020), originally derived from Mouginot et al. (2017) and Rignot et al. (2019). Individual drainage basins (black contours) are shown but are not utilized here. For this study, the following original sectors have been combined: E–Ep+Dp–E, J–Jpp+Jpp–k, and all sectors on the Antarctic Peninsula. Color shadings indicate depth-average (surface-bottom) temperatures (°C) along the Antarctic shelf (poleward of the 1000 m isobath until the ice shelf front) from observations (Section 2.3; Zhou et al., 2024a).

cooling occurs. This creates cold near-surface temperature anomalies and warm anomalies at depth, (Figure 1a). The effects of this redistribution of heat from freshwater "capping" includes the cooling of Southern Hemisphere sea surface and air temperatures (Stouffer et al., 2007; Beadling et al., 2024; Xu et al., 2025; Kaufman et al., 2025), the expansion of Antarctic

sea ice (Beckmann and Goosse, 2003; Hellmer, 2004; Pauling et al., 2016, 2017; Merino et al., 2018; Purich et al., 2018), the reduction of Antarctic Bottom Water (AABW) formation (Fogwill et al., 2015; Lago and England, 2019; Mackie et al., 2020a; Li et al., 2023; Tesdal et al., 2023; Chen et al., 2023), and the warming of the deep ocean (Hansen et al., 2016; Bronselaer et al., 2018; Jeong et al., 2020; Haumann et al., 2020; Moorman et al., 2020; Beadling et al., 2022).

A key question is how these offshore changes interact with the different continental shelf regimes around Antarctica (Thomp-
son et al., 2018), e.g., to what extent stratification-induced deep-ocean warming anomalies propagate to the coast (Thomas et al., 2023), where positive and negative feedbacks have been proposed in response to meltwater input (Bronselaer et al., 2018; Golledge et al., 2019; Snow et al., 2016; Hattermann and Levermann, 2010; Swingedouw et al., 2008). Using a high-resolution ocean model, Moorman et al. (2020) identified two key processes that modulate stratification-induced warming around the Antarctic continent: "isolation" and "homogenization". Isolation occurs as coastal freshening strengthens lateral
density gradients along the Antarctic Slope Front (ASF), reducing interactions between cold shelf waters and warmer open-ocean waters. This process limits the intrusion of offshore deep ocean warming into the continental shelf and can thus lead to local cooling, but with dependencies on the regional ASF dynamics (Hellmer et al., 2017; Hattermann, 2018; Si et al., 2023). Homogenization arises as the strengthened density gradients resulting from coastal freshening accelerate the westward Antarctic Slope Current (ASC) and Antarctic Coastal Current (ACoC), thereby enhancing the lateral connectivity between different
continental shelf regions (Dawson et al., 2023). For example, a stronger ASC and ACoC can transport colder waters from the western Weddell Sea to the warmer Bellingshausen and Amundsen Seas, driving cooling in West Antarctica (Beadling et al.,



2022; Moorman et al., 2020). This westward advection of colder waters does not necessarily result only from strengthened currents; it can also result from a reduction in the formation of Dense Shelf Water (DSW) from the Weddell Sea. Coastal freshening produces slightly lower density DSW that is injected mid-depth rather than sinking and flowing offshore, causing

these lighter waters to become entrained in coastal and slope currents and advected westward around the Antarctic Peninsula (Morrison et al., 2023a). Similar connective links between continental shelf sectors extend along the entire Antarctic margin (Dawson et al., 2023; Beadling, 2023); for example, upstream meltwater advection may also contribute to additional shelf warming in the southern Weddell Sea (Hoffman et al., 2024). Capturing these processes and shelf connectivity depend on the model's ability to represent continental shelf and slope dynamics that govern heat transport between the shelf and the open

ocean, such as localized dense shelf water overflows (Daae et al., 2020; Morrison and et al., 2020), interaction between the ASC and troughs at the continental shelf break (Gómez-Valdivia et al., 2023), episodic atmospheric wind forcing (Morrison et al., 2023b; Dundas et al., 2024), and eddy-driven shoreward transport of CDW (Stewart et al., 2018). Representation of these processes is resolution dependent as the Rossby radius of deformation approaches 1-2 km near the continental shelf (Hallberg, 2013) and a minimum nominal horizontal resolution of 50 km has been shown to be required to resolve robust westward flow

along the slope (Mathiot et al., 2011). For example, a recent study in which a 0.1 Sv meltwater perturbation was imposed identically in two models of differing ocean horizontal grid spacing (GFDL-CM4 and GFDL-ESM4, respectively), showed differing shelf responses (Beadling et al., 2022). The acceleration of the ASC in the (relatively) high-resolution GFDL-CM4 model efficiently isolated the West Antarctic shelf from warming offshore waters, while in the coarser-resolution GFDL-ESM4, a weaker and diffuse ASC allowed meltwater to escape into the open ocean, preventing isolation and causing warming on the continen-

tal shelf in West Antarctica (Beadling et al., 2022). However, to what extent the interplay of stratification-induced warming, cross-shelf isolation, and along-shelf homogenization influences the response to meltwater input in other CMIP-style models remains unknown.

In addition to uncertainties in open ocean–shelf interactions, there is considerable regional and temporal variability and uncertainty about how continental shelf temperature anomalies influence basal melting. Ice shelf basal melting is primarily

governed by the ocean properties beneath the ice and the turbulent processes that transport heat to the ice-ocean interface (Holland and Jenkins, 1999; Rosevear et al., 2025). Since most CMIP-style models do not include ice shelf cavities, basal melt rates in ice sheet models forced by climate models are typically derived using parameterizations that relate melting to ocean thermal forcing extrapolated from "far field" ocean conditions on the continental shelf (Jourdain et al., 2020). Thermal forcing is defined as the difference between the in situ temperature of the ocean and the melting temperature of the ice at the pressure

of the ice shelf base. These parameterizations typically try to account for the modification of ocean properties by the buoyant melt plume along the ice shelf base and subsequent buoyancy-driven circulation in cavities (Jenkins, 1991; Burgard et al., 2022). Parameterizations commonly follow a quadratic function of thermal forcing (Holland, 2008), and approaches such as regional calibrations (Jourdain et al., 2020) and linear response function frameworks (Lambert et al., 2024) have been proposed to account for regional variations in the observed Antarctic melt rates. However, a significant challenge lies in the scarcity of

observational data on oceanographic conditions needed to calibrate these parameterizations. Although basal melt rates can be estimated from remote sensing products (Rignot et al., 2019; Adusumilli et al., 2020; Paolo et al., 2023), direct measurements near or below the ice shelves remain exceedingly rare. As a result, basal melting parameterizations remain poorly constrained, contributing significant uncertainty to estimates of future ice shelf mass loss.

Another key challenge in assessing potential feedback mechanisms is separating the effects of meltwater forcing from

broader changes induced by global warming. Some global warming trends, such as deep ocean warming (Purich and England, 2021), can produce spatial and temporal patterns similar to those driven by increasing meltwater input, making it difficult to isolate individual contributions. Furthermore, global warming alters the hydrological cycle and reduces brine rejection on the continental shelf as sea ice formation reduces, introducing additional freshwater sources that can have impacts similar to





those of Antarctic meltwater discharge (Lockwood et al., 2021). For example, both Goddard et al. (2017); Ong et al. (2024);

Dawson et al. (2025) found ASC responses similar to those reported by Moorman et al. (2020) in response to shelf freshening induced by global warming without additional meltwater forcing. Observational evidence suggests that some of these changes could already be playing out as warming trends in the deep Southern Ocean (Jacobs and Giulivi, 2010; Bintanja et al., 2013; Johnson and Purkey, 2024) suggest potential changes to AABW formation and export processes emanating from Antarctic shelf dynamics. However, it is unclear to what extent the observed trends in bottom water properties are influenced or driven

by on-going changes in meltwater discharge versus other changes in the Southern Ocean freshwater cycle or general warming trends. Earlier simulations that incorporate meltwater forcing based on current Antarctic discharge estimates, combined with high greenhouse gas emission scenarios (RCP8.5), project deep ocean temperature increases between 1°C and 2°C (Bronselaer et al., 2018; Sadai et al., 2020). In these experiments, simulated warming is concentrated in the upper 1000 m and is particularly pronounced along the coasts of the Ross and Weddell Seas, where temperatures are projected to rise by more than 3.5°C and

2.5°C, respectively (Bronselaer et al., 2018). However, it remains uncertain to what extent these subsurface warming patterns are modified by processes identified in standalone meltwater experiments. Specifically, whether climate warming or meltwater input exerts a greater influence on continental shelf properties, or whether meltwater discharge amplifies or counteracts broader global warming-induced changes, also remains an open question (Mackie et al., 2020b).

All of the challenges discussed above are compounded by substantial biases in the ocean properties simulated by climate

models. Climate models contributed to CMIP have been shown to exhibit significant variability in their representation of the water masses of the Southern Ocean (Sallée et al., 2013; Heuzé et al., 2013; Beadling et al., 2019, 2020; Heuzé, 2021), making it difficult to assess whether they realistically capture the conditions necessary to generate accurate ice sheet forcing (Barthel et al., 2020). To mitigate the influence of large mean-state biases, Jourdain et al. (2020) proposes using only anomalies relative to the modern era for ice shelf forcing. However, this approach does not fully resolve the issue, as biases in the mean state

may still influence projected anomalies. This is particularly problematic in meltwater impact studies that rely on a single ocean model, where the results are shaped not only by the model's response to meltwater but also by its inherent mean-state biases. The extent to which model responses depend on their mean state has yet to be fully quantified.

In summary, uncertainties in simulating ice shelf basal melting arise from multiple factors: (a) model- and scenario-dependent climate response and feedback to meltwater discharge, (b) the complex interactions between the open ocean and the continental

shelf, (c) the parameterization of basal melting, which remains poorly constrained due to limited observations, (d) the combined influence of meltwater and broader global warming-induced changes, and (e) biases in climate models' representation of Southern Ocean water mass properties and incomplete representation of near-shelf dynamics. In this study, our objective is to assess all these contributing factors.

We address (a) using a unique set of experiments from a suite of climate models that have contributed simulations to the

novel Southern Ocean Freshwater Input from Antarctica (SOFIA) initiative (Swart et al., 2023). SOFIA provides an experimental framework and coordinated effort that is specifically designed to constrain the climate impacts of additional meltwater associated with Antarctic mass loss and quantify uncertainties from its exclusion in projections. Although following a similar approach to previous hosing experiments, the strength of SOFIA lies in its strict protocol on freshwater timing, magnitude, and distribution, ensuring consistency across simulations. This standardized approach enables direct comparison of model

responses and feedback mechanisms, even across models with different horizontal and vertical resolutions, ocean vertical coordinates, parameterization schemes for subgrid-scale processes, numerics, and underlying biases in the mean state of the models.

Building on the work of Chen et al. (2023), who examined the Southern Ocean deep convection response to Antarctic meltwater in a subset of the SOFIA models, we incorporate additional models that have contributed to SOFIA. We address

(b) by focusing our analysis on the spatial variability in the Antarctic shelf's response to meltwater and whether offshore



**Table 1.** List of models participating in the SOFIA Tier 1 *antwater* experiment used in this analysis. Throughout the manuscript we refer to ACCESS-ESM1-5 as ACCESS-ESM1, AWI-ESM-1-REcoM as AWI-ESM, GISS-E2-1-G as GISS-E2, HadGEM3-GC3.1-LL as HadGEM3 and NorESM2-MM as NorESM2.

| Model | Resolution (ocn/atm, lat×lon, °) | Reference |
|---|---|---|
| ACCESS-ESM1-5 | 1/1.875×1.25 | Ziehn et al. (2020) |
| AWI-ESM-1-REcoM | unstructured 20-100 km/1.865×1.875 | Semmler et al. (2020) |
| CanESM5 | 1/3 | Swart et al. (2019) |
| CESM2 | 1/ 0.9×1.25 | Danabasoglu et al. (2020) |
| EC-Earth3 | 1/1 | Döscher et al. (2022) |
| GFDL-CM4 | 0.25/1 | Held et al. (2019) |
| GFDL-ESM4 | 0.50/1 | Dunne et al. (2020) |
| GISS-E2-1-G | 1×1.25/2×2.5 | Kelley et al. (2020) |
| HadGEM3-GC3.1-LL | 1/1.875×1.25 | Kuhlbrodt et al. (2018) |
| NorESM2-MM | 1/0.9×1.25 | Seland et al. (2020) |

reductions in deep convection induce continental shelf warming. To address (c), we update and calibrate a regional basal melting parameterization, assessing whether shelf warming amplifies ice shelf melting and accelerates Antarctic mass loss. Furthermore, by comparing the ocean response to meltwater with the broader changes induced by global warming in SSP5-8.5 future scenario simulations, we examine (d) whether meltwater is likely to reinforce or counteract warming-driven trends.

Finally, we assess the relationship between model response and biases (e) by incorporating a new high-resolution observational hydrographic climatology. This climatology also aids in refining the basal melting parameterization, providing new insights into the future of Antarctic mass loss.

## 2  Methods

### 2.1  Models and Experimental Design

We use monthly-mean model output obtained from 10 CMIP6 models (Table 1) participating in the *antwater* (Tier 1) experiment described in the SOFIA experimental design (Swart et al., 2023). The models used in this study are the same as those employed in a parallel study (Pauling et al., in prep.), which evaluates the response to sea ice. Following the SOFIA protocol, the *antwater* experiments are branched from the (spun-up) model's *piControl* run. While all other external forcings are kept under the *piControl* conditions, a constant flux of 0.1 Sv (3154 Gt yr$^{-1}$) of additional meltwater is evenly distributed at the

surface across all grid cells adjacent to the Antarctic coast. We note that this amount of meltwater is significantly larger than current observational estimates of basal melt rates on the ice shelf (Adusumilli et al., 2020; Paolo et al., 2023) or Antarctica's current mass imbalance (Slater et al., 2021; **?**). However, the *antwater* experiment is designed to generate a robust signal for model intercomparison, rather than to replicate observed melt rates, and it is not an excessive amount in terms of end-of-21st-century projections (on Climate Change , IPCC). The models employ different vertical and horizontal resolution, but all

implemented extra meltwater at the surface at each model time step. The *antwater* experiment is run for 100 years, and most of the results in our study are presented as anomalies that compare the time averages of the last 10 years *antwater* run against the corresponding time period of the CMIP6 *piControl* run.



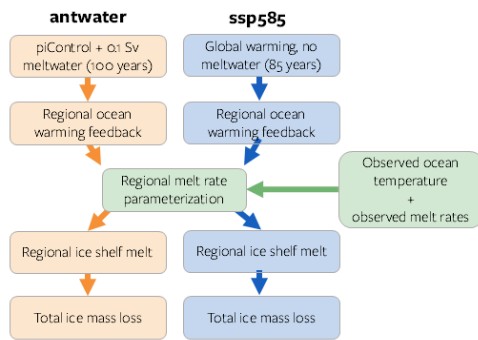

**Figure 2.** Overview of the simulations analyzed in this study and the methodology used to calculate total ice shelf mass loss anomalies based on ocean temperature anomalies.

The SOFIA Tier 2 scenario experiments (Swart et al., 2023) are designed to assess the combined effects of added Antarctic meltwater forcing under future climate conditions. As these simulations are not yet available, we instead compare the *antwater* experiments with future scenario SSP5-8.5 (Meinshausen et al., 2020) anomalies from the CMIP6 ScenarioMIP simulations (O'Neill et al., 2016) to evaluate meltwater effects relative to broader global warming–induced trends. The SSP5-8.5 anomalies represent the change by the end of the 21st century, computed as difference between years 2090–2100 and years 2015–2025. An overview of the different simulations is presented in Figure 2. For each model, only a single run was used, as not all models provided an ensemble. All computations were performed on the models' native grids, except AWI-ESM, whose native grid is unstructured. We calculate model bias as the difference between the model's *piControl* state (last 10 years) and the observationally-based climatology (Section 2.3), with the climatology regridded to match the respective model grids. Here, we use the term bias in a broad sense, as we are comparing preindustrial *piControl* conditions—nominally representing the year 1850—with modern-day observations. While this is not a strict like-for-like comparison, the intent is not to assess the models' skill for a specific historical period. Rather, it is to illustrate the overall offset between the simulated *piControl* state and the observed climate system. We deliberately do not compare with *historical* simulations, as our aim is to evaluate the model state at the branching point of the *antwater* experiments. In addition, future experiments will repeat the same setup using historical climate forcing (Swart et al., 2023). The model output used in this study include ocean potential temperature ('thetao'), salinity ('so'), eastward and northward ocean velocity ('uo' and 'vo'), and the age of seawater since surface contact ('agessc').

Although SSP5-8.5 simulations lack fully interactive ice sheets, they still include additional freshwater inputs from changes in the hydrological cycle, such as altered precipitation, evaporation, continental runoff, and sea ice processes. Precipitation-evaporation (P-E) patterns have already changed (Purich et al., 2018) and are projected to change significantly in the future (Held and Soden, 2006; Bracegirdle et al., 2020; Seroussi et al., 2024), and some models reroute precipitation over the Antarctic continent directly into the ocean as runoff and/or calving. Seasonally and regionally averaged, sea ice growth and melt have a limited impact on the Southern Ocean's net freshwater budget, as most ice grows and melts locally. However, brine rejection remains an important process on the continental shelf (Dawson et al., 2025). To effectively compare the SSP5-8.5 simulations with the *antwater* experiments, it is thus essential to assess the changes in various surface freshwater sources and determine how they compare with the 0.1 Sv anomaly in *antwater*. Figure 3a shows the SSP5-8.5 end-of-the-century anomalies in total freshwater input ('wfo'), evaporation ('evs'), precipitation ('prra'), and river runoff ('friver') integrated over the entire Southern Ocean south of 60°S. ('wfo') includes fluxes associated with calving and sea ice growth and melt, but these are not included in the decomposition. The total freshwater anomaly in the SSP5-8.5 simulations is relatively consistent between models, with





a multi-model mean of 0.14 Sv – slightly higher than the 0.1 Sv added in *antwater*. However, we note that, unlike the constant step forcing in *antwater*, this anomaly evolves over time and is seasonally dependent. In SSP5-8.5, additional surface freshwater is primarily driven by changes in precipitation minus evaporation (P-E; blue bars in Figure 3a), with a smaller contribution from continental runoff (<20% of the anomaly). The resulting surface salinity changes, spatially averaged over the Southern

Ocean, range from -0.15 to -0.55 psu in SSP5-8.5, comparable to the salinity change in *antwater*. We note, however, that the horizontal distribution of the freshwater is very different. In most models, freshwater from runoff and calving is spread out over a much larger area, whereas in *antwater* the freshwater is added adjacent to the coast (See Figure 1 in Kaufman et al., 2025). The purpose of this analysis is to clarify how to interpret the comparison between SSP5-8.5 and *antwater* anomalies. While *antwater* anomalies represent the isolated effect of meltwater alone, SSP5-8.5 anomalies reflect a combination of global

warming-induced changes, including stratification changes due to freshwater anomalies of similar magnitude to *antwater*. However, since SSP5-8.5 does not include additional meltwater forcing, the changes induced by *antwater* could be considered additive to the broader effects observed in SSP5-8.5. Of course, this assumes a linear relationship, which is not entirely accurate (Section 4.2 and 4.4).

### 2.2   Model Output Analysis

Following Jourdain et al. (2020), we assess both offshore deep ocean properties and continental shelf water masses, which are critical for ice shelf–ocean interactions. This approach contrasts with studies that rely on large-scale latitudinal averages (e.g., Levermann et al., 2020; Lambert et al., 2024). To systematically investigate spatial variability, we analyze different regions around Antarctica (Figure 1b), which, for consistency with the Ice Sheet Model Intercomparison Project (ISMIP6; Seroussi et al., 2020), follow the sector definitions of Jourdain et al. (2020). These sectors are based on the latest Ice Sheet Mass Balance

Inter-comparison Exercise (IMBIE) assessment (IMBIE, 2018; **?**) and are delineated using drainage basin boundaries derived from satellite-observed ice sheet surface elevation and velocity data (Mouginot et al., 2017; Rignot et al., 2019). To ensure alignment with coarse model grids and observational data while avoiding the division of similar oceanic "basins", we combine the following sectors: E–Ep+Dp–E, J–Jpp+Jpp–k, and all sectors on the Antarctic Peninsula. Our oceanic sectors are defined by the longitudinal boundaries of the drainage basins, which extend into the open ocean until they reach the continental shelf

break. The boundary between the shelf and the open ocean is marked by the 1000 m isobath, except in the large embayments of the Ross and Weddell seas, where the offshore boundaries follow the definitions of Barthel et al. (2020). We adhere to the IMBIE naming convention but focus specifically on seven key sectors, referred to here by their more commonly recognized names: Weddell, Dronning Maud Land, Amery, Totten, Ross West, Amundsen, and Bellingshausen (Figure 1b).

Consistent with Barthel et al. (2020), who evaluated the output of the CMIP5 models for ice sheet forcing, the oceanic

properties on the continental shelf are calculated as full-depth volume averages (from the surface to the bottom, up to 1000 m depth or less) poleward of the continental shelf break and extending to the "coast". These boundaries vary between models depending on their resolution and bathymetry. We conducted a series of sensitivity tests to ensure that our results are robust with respect to alternative definitions—such as selecting the vertical grid cell closest to the average ice shelf base depth in each region or excluding surface layers. By using the broadest definition (full water column), we reduce the influence of biases

in the vertical distribution of water masses found in some models, and maintain consistency with previous studies. While we acknowledge that temperature responses on the shelf are not vertically uniform, the impact on final results is minor: for example, excluding or including the surface layer alters regional multi-model median temperature anomalies (Section 3.2) by only 0.001°C to 0.028°C, corresponding to relative changes below 7%, with a median absolute difference of just 0.015°C.



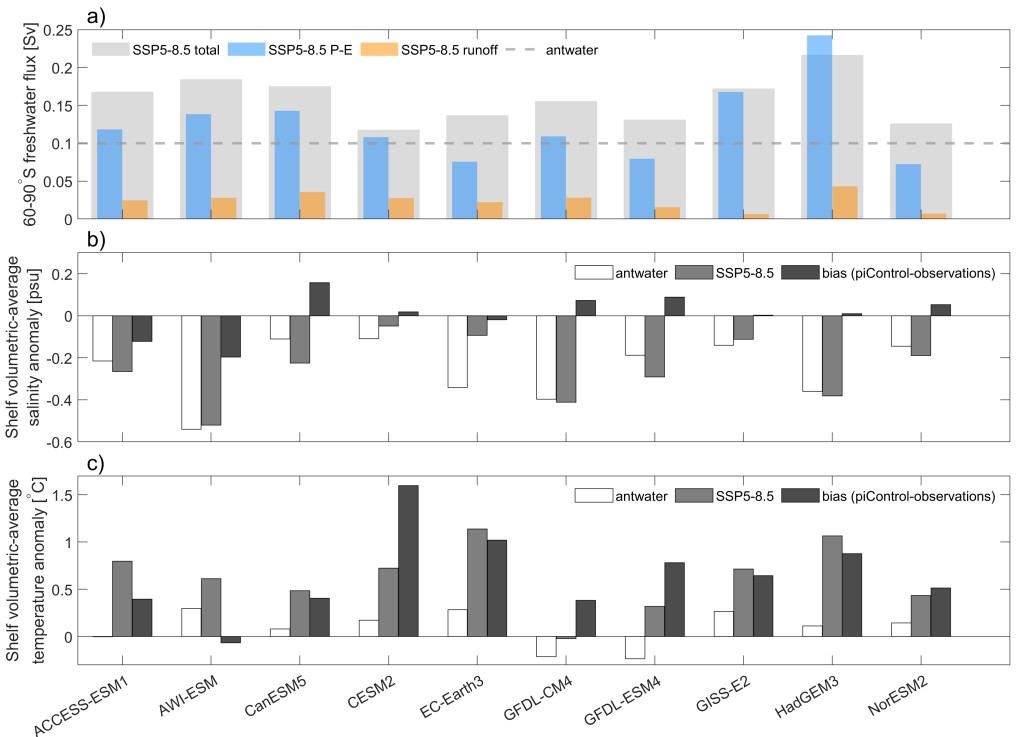

**Figure 3.** a) Magnitude of total freshwater input (Sv) anomaly (shoreward of 60°S) under SSP5-8.5, decomposed into contributions from runoff and precipitation-evaporation (P-E), compared to the 0.1 Sv in the *antwater* experiment. We note that freshwater contributions from sea ice melt/freeze and iceberg calving are not included here, although they do contribute to the total freshwater anomaly shown by the grey bars. Although fluxes related to sea ice melt and freeze are important for seasonal freshwater redistribution, they are of limited relevance on annual timescales in annual means and are therefore examined in detail by Pauling et al. (in prep.). b) Volumetric-average salinity change and bias on the continental shelf (surface-bottom, poleward of the 1000 m isobath until the ice shelf front) in the meltwater perturbation experiments (*antwater - piControl*) and under SSP5-8.5 (comparing 2090–2100 with 2015–2025). c) Same as b) but for temperature.

## 2.3  Contemporary Ocean Climatology

A key limitation in assessing ocean ice shelf interactions is the availability of accurate observational datasets, as many global ocean climatologies exhibit biases in coastal regions around Antarctica. These biases originate from historically sparse observations, particularly on the continental shelf and across the ASF, where strong temperature and salinity gradients are poorly resolved. However, recent observational advances—such as the inclusion of instrumented marine mammals and expanded Argo float deployments—have significantly improved data coverage in these critical regions (Treasure et al., 2017). In this study, we

used a new regional ocean climatology derived from historical CTD, Argo, and seal-borne profiler data (Zhou et al., 2024b), providing a more refined representation of Antarctic coastal ocean properties. The reader is referred to (Zhou et al., 2024a, in



preparation) for full details of the method used to generate the climatology. Here, we briefly introduce the assumption adopted in the creation of ocean climatology.

Assuming anisotropy in water mass properties along ocean currents, the climatology assembles and averages all temperature and salinity profiles within along-flow elongated ellipses. In turn, mean ocean flow is determined using sea surface height (SSH) contours, a proxy for the barotropic geostrophic component of ocean circulation. SSH is provided by the 139th iteration of the Southern Ocean State Estimate (SOSE, Mazloff et al., 2010), which assimilates observations of not only temperature/salinity profiles but also moored time series and surface measurements such as the height of the open ocean sea surface, temperature, salinity and sea ice concentration. The SSH field, therefore, covers the sea-ice area and provides a dynamically conserved estimation of the ocean circulation.

In an attempt to balance sparsity of observations in the Southern Ocean and the need to resolve important spatial gradients in Antarctic shelf seas, the size of the major / minor ellipse axes is further set to be a rough function of the thickness of the water column derived from RTOPO2.0.4 (Schaffer et al., 2016), varying from length scales of 400:200 km in the deep ocean with water column thickness in excess of 4000 m, 268:139 km between 4000 m and 3000 m, and then linearly reduced from 200:100 km to 80:40 km between 3000 m and 1000 m. Circles with a diameter of 20 km are applied everywhere where the water column thickness is lower than 1000 m which we regard as continental shelf regions.

### 2.4 Basal Melting Parameterization

To evaluate the impact of meltwater-induced and global warming-induced changes, we employ a simplified basal melt parameterization to convert simulated coastal ocean temperature anomalies into basal mass loss anomalies of the ice shelf. Following Lambert et al. (2024) and Jourdain et al. (2020), we adopt the proposed (Holland, 2008) quadratic relationship between basal melt rates and regionally averaged thermal forcing: the difference between the simulated ocean temperature and the pressure melting point at ice shelf base–scaled by an effective exchange velocity that relates the bulk properties of the ocean to the heat flux inside the ice shelf cavity (Beckmann and Goosse, 2003). Although this formulation is widely used in ice sheet modeling studies (Favier et al., 2019), Jourdain et al. (2020) highlighted the importance of careful calibration of this parameterization, noting significant regional variability in how efficiently available heat is converted to melting, depending on ice shelf geometry and its influence on melt-induced circulation (Jenkins, 1991; Little et al., 2009).

For each ocean region $j$, the basal melt rate $m_j$ of the associated ice shelf region is calculated as:

$$\dot{m}_j = \gamma_j (T_j - T_f)^2 \tag{1}$$

Here, $\dot{m}_j$ represents the melt rate (m yr$^{-1}$), $T_j$ the volume averaged in situ temperature on the continental shelf in that region (south of the 1000 m isobath), and $T_f$ the in situ freezing temperature at the ice shelf base depth in the same region. The parameter $\gamma_j$ (m yr$^{-1}$ °C$^{-2}$) is the local melt rate coefficient that varies across Antarctica. Using the same framework as Jourdain et al. (2020) but incorporating updated datasets - specifically, the new ocean climatology and the most recent basal melt rates derived from satellites (described in the following section) - we derive regionally calibrated $\gamma_j$ values.

For each region $j$, we extract the thermal climatological forcing, defined as the difference between the observed ocean temperatures on the continental shelf ($T_j^{\mathrm{obs}}$) and the freezing temperature at the ice shelf base depth ($T_f^{\mathrm{obs}}$), which is determined by the climatological salinity. The regional melt rate coefficient $\gamma_j$ is then calculated as:

$$\gamma_j = \frac{\dot{m}_j^{\mathrm{obs}}}{(T_j^{\mathrm{obs}} - T_f^{\mathrm{obs}})^2} \tag{2}$$



### 2.5 Observed Melt Rates and Ice Shelf Draft

Our basal melting parameterization is limited by the updated dataset of satellite-derived basal melting of Antarctic ice shelves
from Paolo et al. (2023), which combines observations from four satellite radar altimeters to produce a 26-year (1992–2017)
pan-Antarctic time series of ice shelf thickness and basal melt rates at 3 km spatial resolution. Previous Antarctica-wide
estimates were limited to shorter time periods (Depoorter et al., 2013; Rignot et al., 2013) or offered lower spatial resolution
over longer durations (Adusumilli et al., 2020), while this dataset combines enhanced temporal coverage and finer spatial
resolution. Paolo et al. (2023) report a temporal mean basal melt rate of $965 \pm 265\,\mathrm{Gt\,yr^{-1}}$ over 1992–2017, which aligns well
with the $1325 \pm 235\,\mathrm{Gt\,yr^{-1}}$ rate for the 2000s reported by Rignot et al. (2013). For our purpose, we calculate the average melt
rates for the entire observation period as area-weighted means for each sector (Table 2). To calculate the difference from the
freezing temperature at the ice shelf base, we need an observationally based estimate of the ice shelf draft. For this purpose,
we use the high-resolution ice shelf draft dataset developed by Moholdt and Maton (2024), which is averaged over the same
sectors shown in Figure 1b.

### 3 Results

#### 3.1 Regionally Varying Meltwater-induced Subsurface Warming

As previously shown by Chen et al. (2023), the zonal mean temperature anomaly across the different models reveals distinct
patterns of warming and cooling in the Southern Ocean in response to the *antwater* meltwater perturbations (Figure 4). All
models consistently show a cooling response at the surface and warming in the deeper ocean layers, which aligns with the
expected stratification-induced effects of meltwater. For some models, the warming anomaly extends to the surface near the
continent, but for most models, the maximum anomaly is below 1500 m. In general, the response is strongest around and south
of 65°S latitude, corresponding to key regions of modelled (impeded) deep water formation (Heuzé, 2021, show that CMIP6
models typically fail to produce dense water in the same regions where it forms in the real world). There is a wide range of
warming magnitudes, where models such as AWI-ESM, GFDL-CM4 and GISS-E2 exhibit pronounced warming ($> 1°$ C),
while others such as ACCESS-ESM1 and NorESM2 show relatively subdued warming. No clear relationship between model
resolution and the simulated response is evident (all models are considered "coarse" except GFDL-CM4 and AWI-ESM).
Above the subsurface warming, models consistently exhibit surface cooling (consistent with results from Kaufman et al.,
2025), though its intensity and vertical extent vary across models. The strongest and deepest cooling occurs north of 55°S, in
regions where Subantarctic Mode Water and Antarctic Intermediate Water are known to subduct in the mean state. The cooling
patterns also correspond to the underlying climatological mean temperature. This cooling anomaly promotes the expansion
of sea ice, as shown in a parallel study (Pauling et al., in prep.), which also links the extent of surface cooling to mean-state
stratification—consistent with our finding that models exhibiting strong surface cooling tend to show pronounced warming at
depth.

At the continental shelf break and on the continental shelf, there are pronounced inter-model differences in both the sign
and magnitude of the temperature anomalies resulting from the *antwater* experiment. Likewise, models differ in the steepness
of isotherms in the climatological mean near the shelf break. The zonal mean, volume-averaged temperature anomaly over the
continental shelf (from surface to bottom, poleward of the 1000m isobath) is generally positive across most models (as also
shown in Figure 3c), but notably negative in both GFDL models, discussed further below.

Although the zonal mean provides a broad overview of the temperature response to meltwater, previous studies (Beadling
et al., 2022) have shown strong regional variations, and thus a zonal mean is less appropriate to study changes along the
continental shelf, requiring us to examine how these responses differ between key sectors.



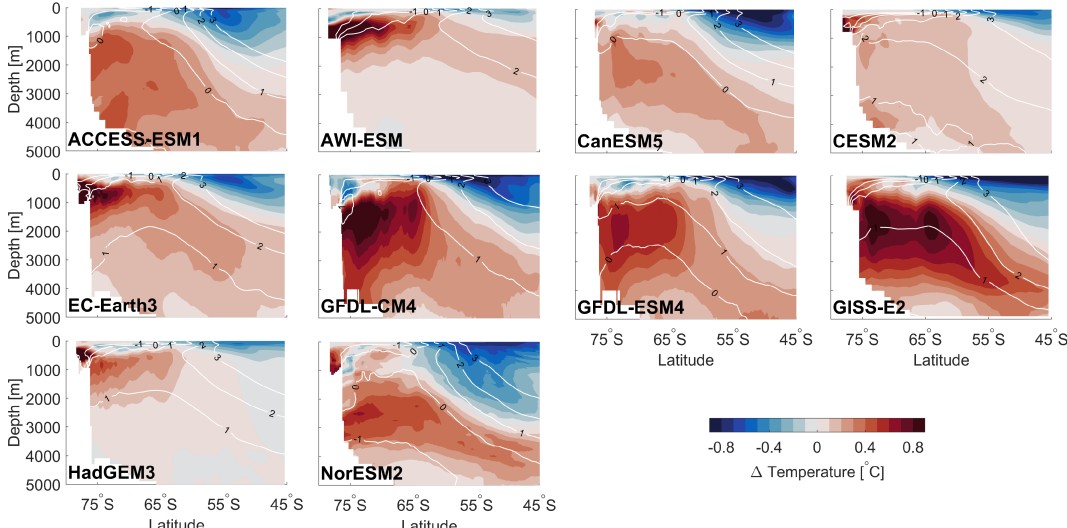

**Figure 4.** Zonal mean temperature change in response to meltwater perturbation (°C; *antwater - piControl*; last 10 years) for all models. White contours represent the climatological mean temperature in *piControl*.

Figure 5 and 6 present the spatial distribution of *antwater* anomalies in relation to model biases and the temperature anomalies projected under the SSP5-8.5 global warming scenario. In particular, the temperature biases in most models (first column in Figure 5 and 6; *piControl* minus observations) are larger than the anomalies induced by the *antwater* experiment (second column). The biases exceed 1°C in both positive and negative directions, displaying pronounced differences among models. It is important to note, however, that some degree of cool bias is expected, since the *piControl* simulations represent pre-industrial conditions, while the observations reflect a climate that has already experienced over a century of global warming. Consequently, warm biases in the models suggest they may be substantially too warm. Some models, such as CESM2, EC-Earth3, GISS-E2, and HadGEM3, show predominantly warm biases across all regions surrounding the Antarctic continent. In contrast, ACCESS-ESM1 and NorESM2 are predominantly biased cold, except in regions like West Antarctica and the Totten and Amery sectors. CanESM5 and both GFDL models show a distinct pattern of warm biases along the coast, transitioning to cold biases farther offshore.

Temperature biases are especially large on the continental shelf. Near the coast, most models exhibit warm biases, particularly in West Antarctica, with AWI-ESM being the only model that shows cold biases in this region. Salinity biases also tend to be large near the coast (fourth column in Figure 5 and 6). We note, however, that the observational data near the coast are also uncertain and may be affected by undersampling of internal variability. These uncertainties are discussed further in Section 4.4. Although most models exhibit a fresh bias across the open Southern Ocean (a well-known bias in CMIP6 models, Purich and England (2021), shared with the CMIP5 predecessors Beadling et al. (2019)), coastal regions, particularly in West Antarctica, tend to be too saline, pointing to a bias in the properties, residence time or amount of CDW on the shelf. An exception of the too saline shelves is the southern Weddell Sea, where most models show a fresh bias even on the shelf, likely due to the absence of wind-driven coastal polynyas that facilitate the formation of sea ice and the densification of the shelf water masses (Vernet



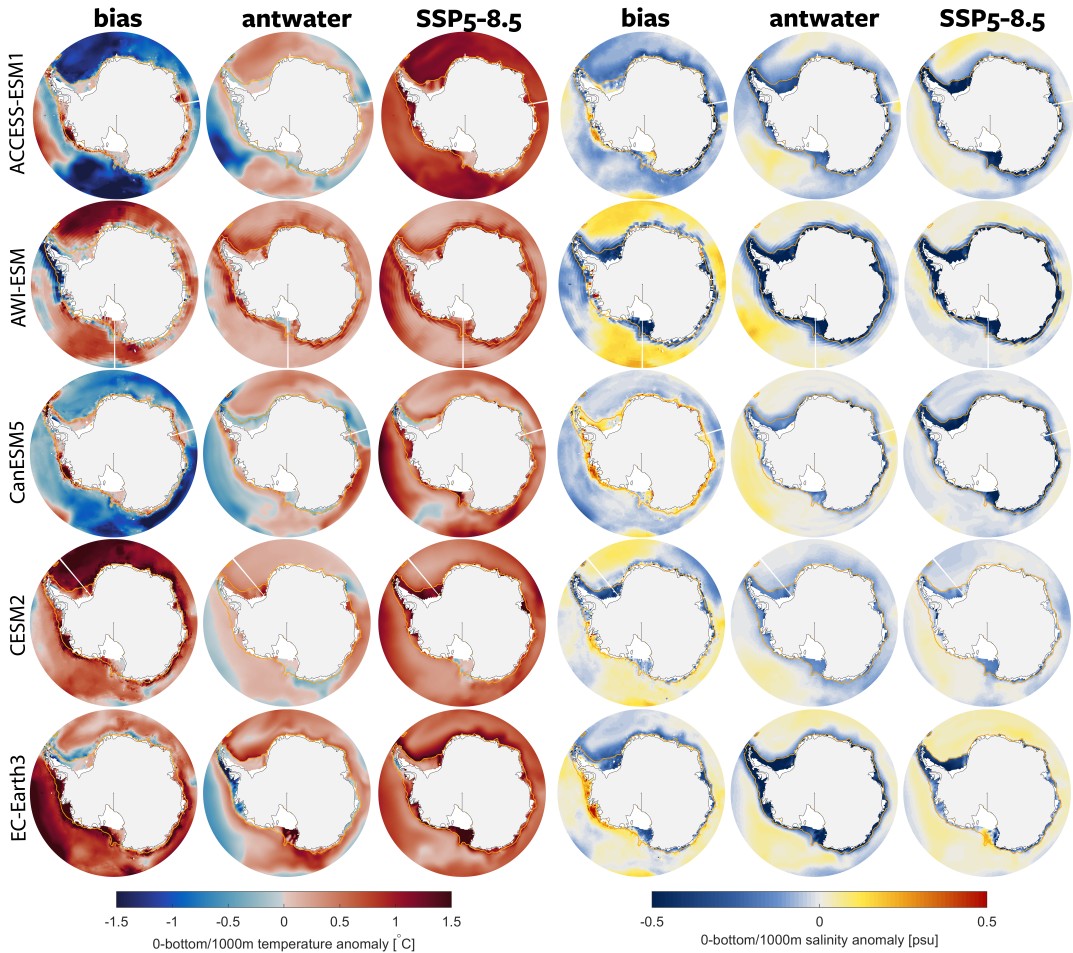

**Figure 5.** Spatial maps of surface–1000 m/bottom depth-averaged model temperature (°C) and salinity. The first column shows temperature bias (*piControl–observations*), the second column shows temperature change due to meltwater perturbation (*antwater–piControl*, comparing the last 10 years), and the third column shows temperature change under SSP5-8.5 (comparing 2090–2100 with 2015–2025). Columns 4–6 display the corresponding salinity maps. The 1000 m isobath, marking the boundary between the continental shelf and the open ocean, is contoured in each panel.

et al., 2019). Model resolution is likely a key factor; for example, the relatively high-resolution GFDL models form dense shelf water in the Weddell Sea via coastal polynyas in their mean state (Tesdal et al., 2023) and exhibit smaller salinity biases in this region. Nonetheless, large biases in continental shelf properties are expected, as these regions are inherently more challenging to resolve in models. Accurate representation requires capturing cross-front exchanges at the shelf break and coastal water mass transformation processes, which themselves depend on mesoscale eddy processes (Hallberg, 2013). Moreover, we may suspect a greater uncertainty in the climatology in these sparsely observed coastal regions. Notable indications of the latter include excessively low temperatures in the warm shelf region of the Amundsen Sea, and overly warm conditions along the narrow continental shelf of fresh shelf regions, such as the coast of Dronning Maud Land, as both regions show comparable



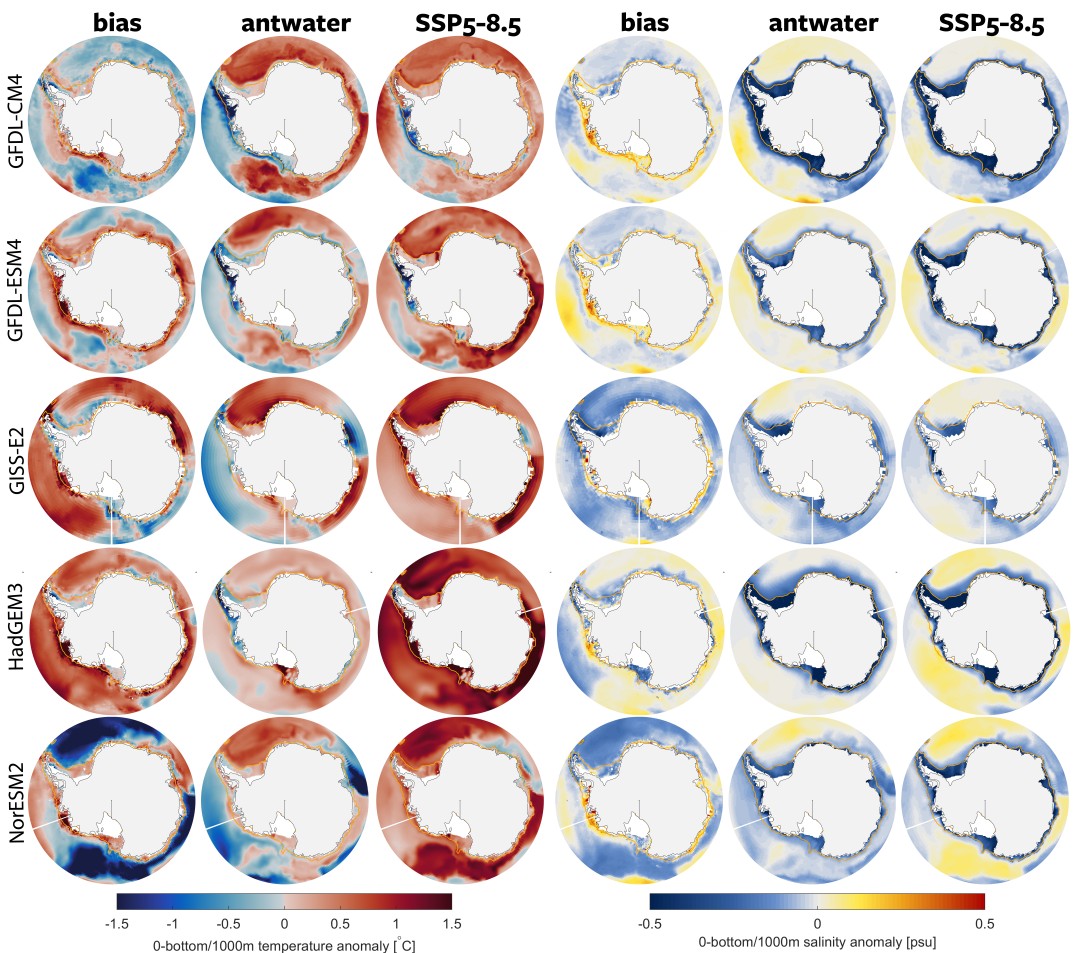

**Figure 6.** Continuation of the spatial anomaly maps shown in Figure 5.

thermal forcing in our analysis (Figure 10b). However, in other regions, the climatology aligns well with previous studies, such as in the Bellingshausen region (Schubert et al., 2021).

Spatial patterns in response to the *antwater* meltwater simulations are remarkably consistent in all models, showing greater agreement than mean-state biases. Generally, a meltwater perturbation induces an offshore warming anomaly in the upper 1000 m, but this warming is not uniform across all regions. The maps illustrate that the offshore circumpolar mean warming shown in Figures 4 is primarily driven by warming in the Weddell Sea and along the east Antarctic coast, while an offshore cooling anomaly is evident in most models west of the Antarctic Peninsula. These patterns underscore substantial regional differences, demonstrating that a zonal mean alone is insufficient to represent changes in the Southern Ocean. As expected, the SSP5-8.5 simulations show a widespread warming in all models (fourth column in Figure 5 and 6). However, regional differences are also substantial in the SSP5-8.5 simulations, and several models also exhibit weaker warming or even cooling anomalies west of the Antarctic Peninsula. This asymmetric response and the potential mechanisms driving the cooling will be addressed in detail in Sections 3.2 and 4.3. It is also important to note that, even when averaging over multiple years,



natural variability may still contribute to some of the spatial differences seen in the anomaly maps (Purich and England, 2021), as many of these models exhibit substantial multi-decadal and centennial-scale variability (see Supplementary Material in Beadling et al., 2020).

In the *antwater* simulations, the models vary significantly in whether the warm offshore anomalies in the Weddell Sea and East Antarctica extend southward past the continental shelf break. In some models, the warming response is equally strong near the coast as offshore, while in others, the magnitude of coastal anomalies is somewhat reduced. In the SSP5-8.5 simulations, the most pronounced temperature increases occur in the open ocean, whereas coastal regions, particularly near the shelf break, exhibit more moderate warming. The Amundsen and Bellingshausen regions are exceptions, showing little or no gradient

across the shelf break in both *antwater* and SSP5-8.5, consistent with the direct isopycnal connection characteristic of this shelf regime (Thompson et al., 2018). Although the spatial patterns of SSP5-8.5 anomalies are relatively consistent across models, there is considerable variation in magnitude. The strongest warming near the coast is observed in HadGEM3, EC-Earth3, CESM2, and CanESM5. This contrasts with the *antwater* simulations, where the strongest coastal warming anomalies occur in GFDL-CM4 and AWI-ESM1.

Similar to the temperature response, models exhibit substantial variation in the northward extent of negative salinity anomalies: in some cases, freshening is confined to the immediate coastal zone, while in others it extends farther offshore. GFDL-CM4 and AWI-ESM also show the strongest salinity response near the coast (see also Figure 3b). Interestingly, these are the models with the highest spatial resolution along the coast. This is consistent with the findings of Beadling et al. (2022), who suggested that coarse-resolution models, due to a poorly resolved ASC, may allow meltwater to escape into the open ocean, thereby

limiting shelf isolation and warming.

The spatial patterns of salinity anomalies are relatively similar between the SSP5-8.5 and *antwater* simulations, but arise from different mechanisms. Reflecting the prominent influence of P-E, which is not limited to the coastline, the negative salinity anomalies expand further north in most models under SSP5-8.5. Furthermore, reduced sea ice formation in SSP5-8.5 (Roach et al., 2020) leads to reduced brine rejection, further contributing to negative salinity anomalies. The surface salinity changes

in *antwater* are not investigated here, but are discussed in detail in Pauling et al. (in prep).

### 3.2  Warming or Cooling on the Continental Shelf

Figure 5 and 6 highlight the nonuniformity of subsurface warming responses around Antarctica. A key question for assessing potential melting feedbacks is whether offshore anomalies propagate onto the continental shelf and how they influence temperatures at the ice shelf front. To investigate this, Figures 7 and 8 present the volume-averaged continental shelf temperature

anomalies from both the *antwater* and SSP5-8.5 simulations as a function of longitude. Notably, these results differ markedly from the zonal mean volume-averaged anomalies shown in Figure 3c, highlighting the importance of regional variability. Taking into account the varying magnitude of the anomalies in different models (different scaling of the y axis in 7 a), a consistent pattern emerges in the *antwater* simulations: warming anomalies dominate the Weddell Sea and much of the East Antarctic shelf, while cooling anomalies appear west of the Antarctic Peninsula in 6/10 models with additional two models show weaker

than zonally averaged warming. Warming anomalies typically reach up to 0.5 °C, while cooling anomalies range from -0.5 °C to as low as -1.5 °C, with the strongest cooling observed in the GFDL models. In contrast, the SSP5-8.5 simulations exhibit a more uniform warming signal along the Antarctic shelf. The SSP5-8.5 simulations show widespread and consistent warming across the shelf in most models, with larger anomalies than those induced by the additional meltwater forcing in *antwater* (Figure 8). Shelf temperature anomalies under SSP5-8.5 typically reach up to 1 °C, reflecting the cumulative effects of long-term

global warming. AWI-ESM is the only model that does not exhibit a cooling response in any region around the continent.

The multi-model spread and median of continental shelf temperature anomalies across the regions defined in Figure 1b are summarized in Figure 8. Following the interquartile range, there is significant warming in response to *antwater* in Dronning



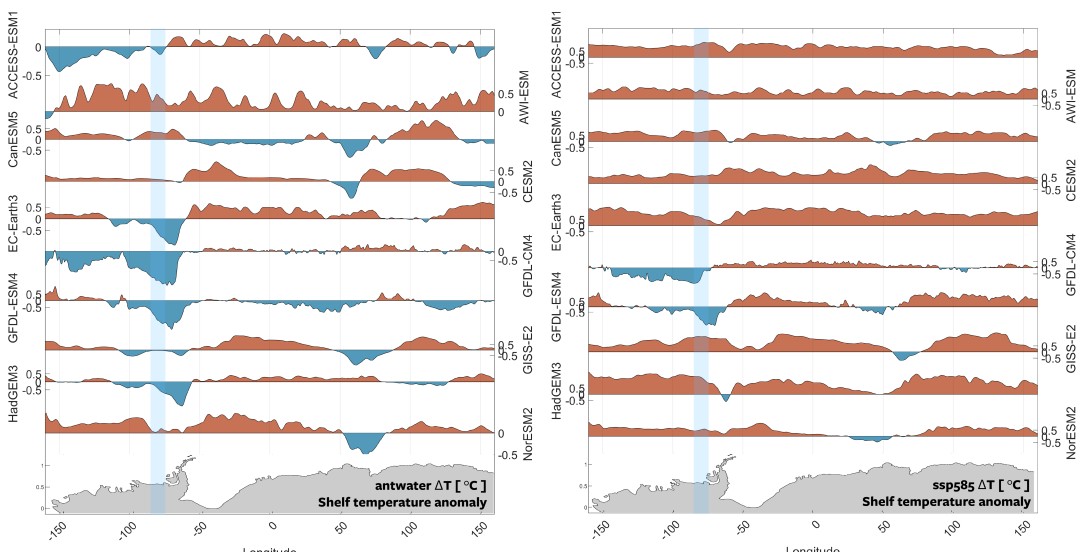

**Figure 7.** Circum-Antarctic mean shelf temperature change (0-bottom, poleward of the 1000 m isobath until the ice shelf front) in response to meltwater perturbation (*antwater–piControl*, comparing the last 10 years; left) and under SSP5-8.5 (comparing 2090–2100 with 2015–2025; right) for all models. Note the different scales on the y-axis for the different models. Shaded regions indicate the Bellingshausen sector, where multiple models show negative anomalies, which are examined in detail later in the paper.

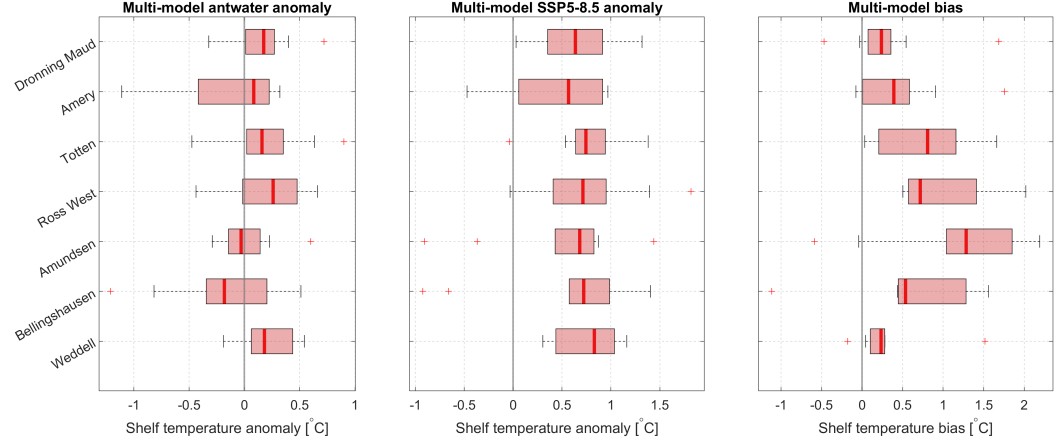

**Figure 8.** Box plots of shelf (0-bottom, poleward of the 1000 m isobath to the ice shelf front) temperature anomalies for all models combined in the regions defined in Figure 1. The rectangle represents the second and third quartiles of all models, the red line indicates the multi-model median value, and the whiskers show the maximum model spread. Red plus signs indicate outliers (values beyond 1.5 times the interquartile range from the quartiles). The left column shows the multi-model temperature change due to meltwater perturbation (*antwater - piControl*, comparing the last 10 years), the second column illustrates the multi-model temperature change under SSP5-8.5 (comparing 2090–2100 with 2015–2025), and the third column shows the multi-model bias (*piControl - observations*).





Maud Land, Totten, Ross West and Weddell, while Amery, Amundsen and Bellingshausen are inconclusive. All show signifi-
cant warming in SSP5-8.5. The Dronning Maud Land, Ross West, Totten, and Weddell regions show considerable multi-model
median warming under *antwater* (0.17 °C, 0.26 °C, 0.16 °C, and 0.18 °C, respectively), with even stronger warming under
SSP5-8.5 (0.64 °C, 0.65 °C, 0.66 °C, and 0.82 °C, respectively). In contrast, the Amundsen and Bellingshausen regions ex-
hibit cooling in the *antwater* multi-model median. The Amery region is also notable for showing cooling or subdued warming
anomalies in several models. Under SSP5-8.5, most models indicate warming, but both GFDL models still show cooling in the
Amundsen and Bellingshausen regions, although other studies suggested that this signal may partly reflect internal variability
within the model (Purich and England, 2021).

The cold anomaly in the Amundsen and Bellingshausen regions in response to meltwater was previously reported by Bead-
ling et al. (2022) for the GFDL models in response to experiments similar to those studied here (0.1 Sv non-spatially-uniform
forcing), but we now show that this feature is evident in 6/10 of the SOFIA simulations. The mechanisms driving this cooling
west of the Antarctic Peninsula are explored in Section 4.3, but in particular these regional patterns do not align with the
*piControl* temperature distribution, which features the warmest waters in West Antarctica (not shown). Among models that
do not show a distinct cooling anomaly in West Antarctica, CESM2 and NorESM2 exhibit reduced warming, suggesting that
similar mechanisms may be at play, but to a lesser extent. Furthermore, both GFDL models show a cooling anomaly in West
Antarctica under SSP5-8.5, implying that comparable processes may be influencing both experiments. However, in SSP5-8.5,
the signal may be weaker, masked, or offset by the general warming of the Southern Ocean under global warming scenarios.

The Amery region also exhibits a cold anomaly or, at minimum, a subdued warming response in several models under both
*antwater* and SSP5-8.5 simulations, though the multimodel median remains positive. In models in which a strong cooling
anomaly is present west of the Antarctic Peninsula, the Amery response is generally weaker. In NorESM2, GISS-E2, CESM2,
and CanESM5, the cooling anomaly in the Amery region is more pronounced than that west of the Peninsula.

No consistent relationship is found between the magnitude of continental shelf temperature biases and the magnitude or sign
of the resulting shelf temperature anomalies. In all regions, median temperature biases exceed the median *antwater* anomalies.
These biases range from 0.2 °C in the Weddell region to 1.2 °C in the Amundsen region.

The intermodel spread varies considerably across regions, with the Bellingshausen region standing out as having the largest
overall spread. The pronounced intermodel variability, along with the distinct cold anomalies seen in some models, highlights
the need for further investigation of this region. Figure 9 presents vertical sections of all models on the Bellingshausen shelf,
illustrating the vertical and horizontal extent of these anomalies and their relationship to model biases in this region.

The Bellingshausen continental shelf is classified as a "warm" shelf regime (Thompson et al., 2018), characterized by the
presence of a relatively warm and salty unmodified CDW on the continental shelf, and the absence of a pronounced ASF. Most
models capture this direct isopycnal connection between the shelf and the deep ocean (first column, Figure 9) and simulate
warm waters reaching the coast. However, AWI-ESM stands out as the only model without a warm shelf regime here, consistent
with its unique response to *antwater* compared to the other models (second column, Figure 9). Although offshore temperature
structures and mean states are relatively consistent across models, they differ in the southward extent of CDW intrusion and the
steepness of temperature gradients near the shelf break. Some models exhibit a distinct slope front, while others display flatter
isotherms extending southward. The biases are generally the largest on the shelf and the lowest offshore, with most models
showing a warm bias compared to the climatology in this region.

Models that exhibit a clear on-shelf cold anomaly in response to *antwater* typically show this anomaly extending from 50 m
depth to the shelf bottom, spanning from the shelf break to the continental interior. An exception is EC-Earth, which shows
a cold anomaly through most of the water column, but warming near the bottom on the shelf. NorESM2-MM also shows a
cold anomaly off-shore, but this is counteracted by a warm anomaly just on the shelf, coinciding with steep gradients and
cold biases in the *piControl* — potentially reflecting a frontal current with anomalous properties. AWI-ESM also shows a



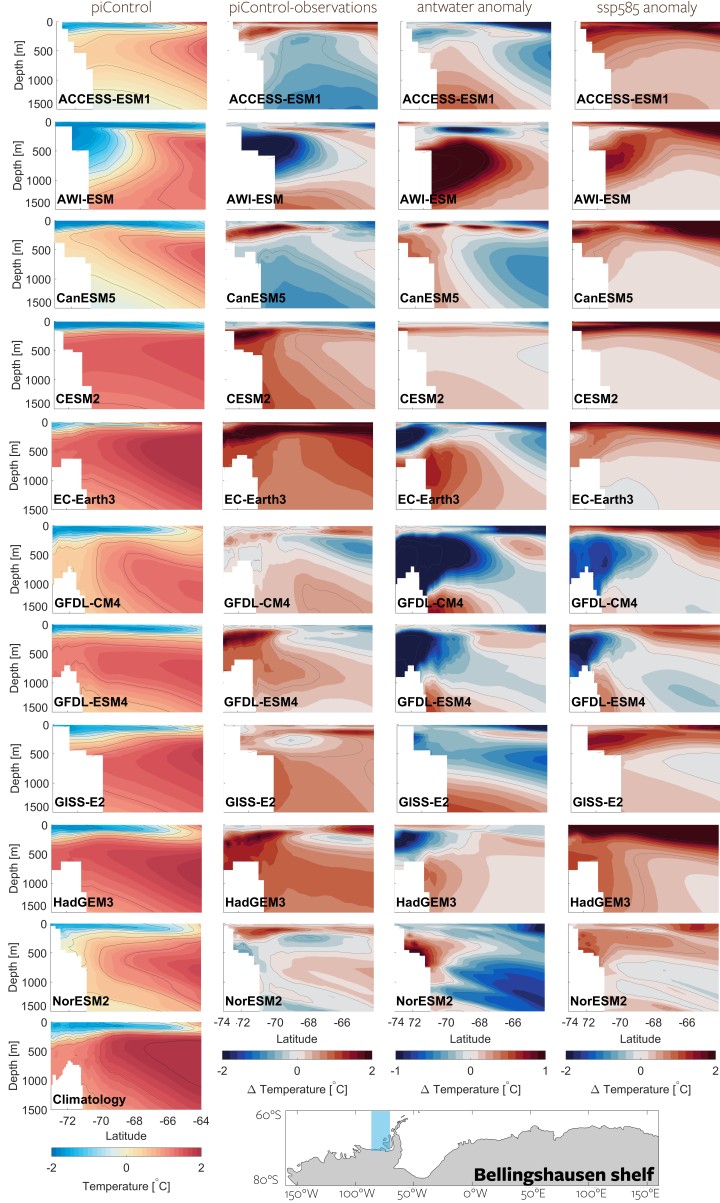

**Figure 9.** Example transect of the Bellingshausen region, where the shelf temperature response to an additional meltwater forcing is negative, contrasting with other regions. The first column presents the model's climatological mean temperature in the upper 1500 m, extending northward from the continental shelf. The second column shows the bias (*piControl - observations*), the third column displays the temperature change due to meltwater perturbation (*antwater - piControl*, comparing the last 10 years), and the fourth column illustrates the temperature change under SSP5-8.5 (comparing 2090–2100 with 2015–2025). Note that all data are plotted on the model's native grids, displaying varying horizontal and vertical resolutions and shelf extents (except AWI-ESM, whose native grid is unstructured). However, detailed bathymetry from each model is not shown. The observational climatological mean is shown at the bottom.





cold anomaly, but it is limited to a thin shallow layer, and hence, deeper subsurface warming dominates the shelf response. CESM2 is the only model with no cold anomaly in the Bellingshausen region. Under SSP5-8.5, all models show consistent shelf warming mainly confined to the upper 500 m, consistent with surface warming under global climate change. However, the Bellingshausen region exhibits less pronounced warming compared to other regions, with GFDL models even maintaining cooling on the shelf. This suggests that the mechanisms driving the cooling response in *antwater* may also influence the shelf

response in SSP5-8.5, although to a lesser degree (also discussed in Section 4.3).

In general, the Bellingshausen region demonstrates inconsistent *antwater* anomalies despite relatively similar biases, compared to regions such as Dronning Maud Land, Totten and Weddell, where consistent *antwater* responses occur despite varied biases.

### 3.3 Present Day Melt Rates and Regional Parameterizations

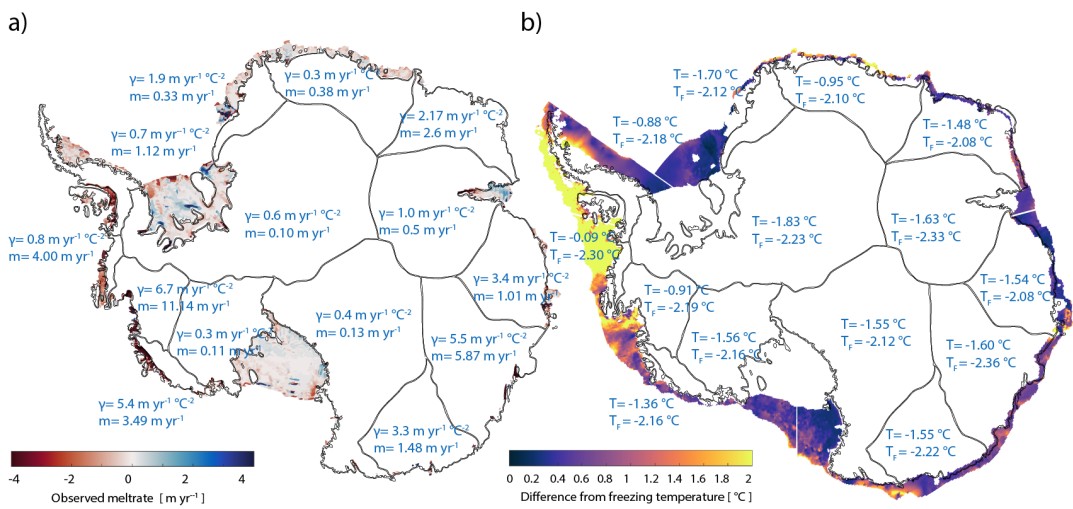

**Figure 10.** a) Observed 1992–2017 mean basal melt rates (from Paolo et al., 2023) shown in color shading. Values represent regional mean melt rates and the $\gamma$ coefficient, which is derived using the local quadratic melting parameterization, as described in equation 2. b) Shelf thermal forcing (defined as the temperature above the local freezing point; colorbar), along with regional mean values of shelf temperature ($T$) and freezing temperature ($Tf$) at the depth of the ice shelf base (based on the spatially averaged ice shelf draft; see Section 2.5). Regions are defined as areas poleward of the 1000 m isobath extending to the ice shelf front. Temperature data are from observations (Section 2.3; Zhou et al., 2024a).

Following the approach of Jourdain et al. (2020), we translate simulated ocean temperature anomalies into basal mass loss anomalies, as detailed in Section 2.4. Due to significant biases in the mean temperatures modeled around the Antarctic coast (Section 3.1), we use updated observationally-based climatology and updated satellite-derived basal melt rate estimates to calibrate the basal melt parameterization. Satellite-derived temporal mean basal melt rates for 1992–2017 are presented in Figure 10a, with values representing the spatial mean for each IMBIE region. Area-averaged present-day melt rates range from

0.10 m yr$^{-1}$ in the Weddell region to 11.14 m yr$^{-1}$ and 5.87 m yr$^{-1}$ in the Bellingshausen and Totten regions, respectively.

Figure 10b presents the temporally averaged thermal forcing (see Methods), calculated from observational climatology, across the continental shelf from the 1000-m isobath to the ice shelf fronts (except in the Weddell and Ross regions where the northern boundary has been moved southward to account for the large shelf area). The values reflect the regional mean shelf





temperatures ($T$) and the reference melting point temperatures ($T_f$) at the respective mean ice shelf drafts (Section 2.5). Thermal forcing is highest in the warm regions of West Antarctica, reaching up to 2.21 °C, while it remains relatively low in East Antarctica, typically ranging between 0.57 °C and 1.15 °C. Despite incorporating more recent data than other climatological datasets, the climatology remains subject to uncertainty and potential subsampling of temporal variability (discussed further in Section 4.4). For example, the estimated shelf temperatures in Dronning Maud Land ($T = -0.95°$ C) are probably overestimated compared to Hattermann et al. (2012); Lauber et al. (2024), whereas they are likely underestimated in the Amundsen Sea ($T = -0.91°$ C) compared to the observational estimates from Jenkins et al. (2010); Nakayama et al. (2019). Nonetheless, we use this temperature dataset throughout the analysis to ensure consistency across regions.

The local melt rate coefficient, $\gamma_j$, is calculated for each region (Equation 2) to quantify the sensitivity of the melt rates to thermal forcing. The $\gamma_j$ values (Figure 10a) span a wide range, from 0.3 m yr$^{-1}$ °C$^{-2}$ in the Ross and Dronning Maud Land regions to 6.7 m yr$^{-1}$ °C$^{-2}$ in the Bellingshausen region. The spatial average $\gamma_j$ is 2.1 m yr$^{-1}$ °C$^{-2}$, closely aligning with the spatial mean of 2.2 m yr$^{-1}$ °C$^{-2}$ suggested by Jourdain et al. (2020). These results highlight the considerable spatial variability in melt sensitivity, while also reflecting substantial uncertainty. Specifically, this uncertainty arises from the potential overestimation of thermal forcing in climatology and the inherent limitations in the formulation of melt rate parameterizations (Burgard et al., 2022).

### 3.4 Future Melt Rates and Projected ice shelf mass loss

To assess the impact of model biases on melt rates, we first calculate the mean melt rate using the uncorrected bias *piControl* model temperatures for our selected regions (top row in Figure 11). In general, melt rates derived from *piControl* are overestimated by a factor of 2 to 6 compared to present-day observations. However, in some regions—such as Dronning Maud Land and the Weddell Sea—the agreement is notably better. These findings underscore the need for bias correction or the use of temperature anomalies alone, as recommended by Jourdain et al. (2020), when employing climate model temperature fields to estimate basal melt rates. It is important to note that *piControl* and observational estimates cover different time periods; the absence of anthropogenic warming in *piControl* should lead to underestimated melt rates. Future experiments will repeat this setup under historical and future scenario forcing (Swart et al., 2023). Nevertheless, the regional pattern in calculated melt rates corresponds well with observations—regions with low observed melt also simulate low melt rates, and vice versa (noting the different y-axis scales)—indicating that the regionally dependent $\gamma$ parameter performs as intended. Lastly, we note that salinity variations have a negligible impact on freezing temperature in this context, with pressure and temperature exerting far greater influence.

We now explore the melt rate anomalies resulting from the *antwater* and SSP5-8.5 simulations, shown in the lower panels of Figure 11. Figure 11 illustrates both the anomalous melt rates (left panels) and the corresponding regional mass loss (right panels), obtained by scaling the melt rate anomalies by the total ice shelf area in each region. As anticipated, the difference in the melting anomalies between *antwater* and SSP5-8.5 closely mirrors the pattern of their respective warming anomalies, with the highest melting observed under SSP5-8.5. Negative temperature anomalies correspond to negative melting anomalies, underscoring the direct link between ocean thermal forcing and basal melting. For *antwater*, the multi-model median melt rate anomalies range from -0.8 m yr$^{-1}$ in the Bellingshausen region to +2 m yr$^{-1}$ in the Totten region, but typically with an order of magnitude greater variability between different models. Under SSP5-8.5, the multimodel median melt rate anomalies increase to vary from +0.5 m yr$^{-1}$ in the Dronning Maud Land region to +28 m yr$^{-1}$ in the Amundsen region. The basal melt rates of the mean state are overestimated (top panels, Figure 11), likely due to warm biases in the simulations of the model (Figure 8). However, it is also possible that the parameterization itself is overly sensitive, contributing to the overestimation. If this is the case, then the melt rates derived from the temperature anomalies (lower panels, Figure 11) may also be overestimated.



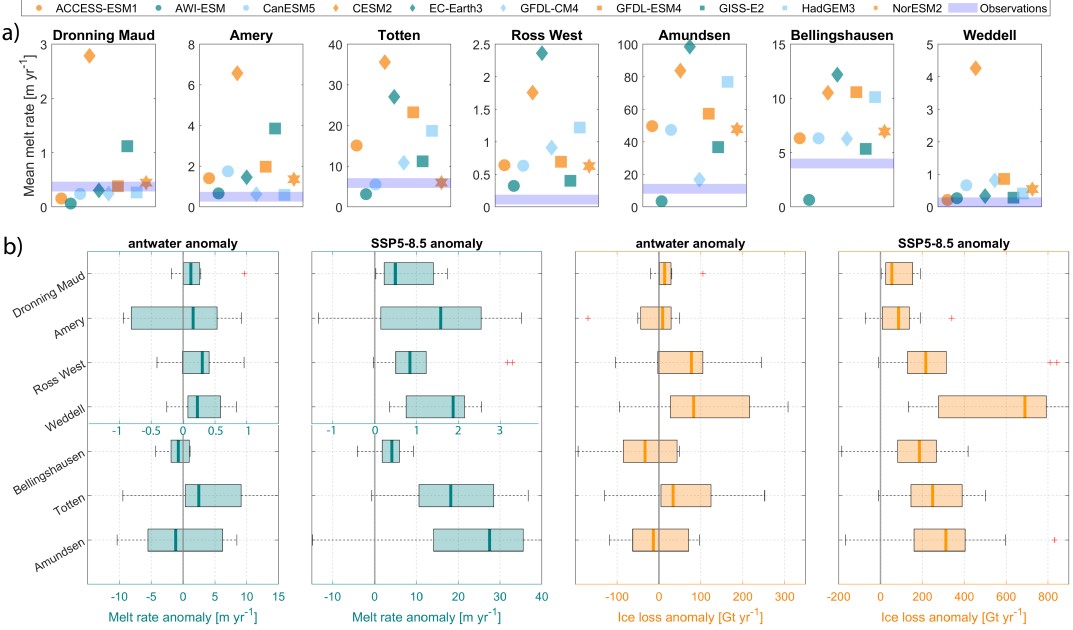

**Figure 11.** a) Simulated mean melt rates (*piControl*) calculated using Eq. 1 and 2 with simulated ocean temperatures and $\gamma$ values from Figure 10, compared with observed melt rates for regions defined in Figure 1. b) Box plots of multi-model simulated melt rate anomalies (left panels) and total ice shelf mass anomaly due to basal melt (right panels) for the same regions as in a), based on meltwater experiments and under SSP5-8.5. Note different x-axis scales for the melt rates in the Amundsen, Totten, and Bellingshausen regions. The rectangle represents the second and third quartiles of all models, the thick colored line indicates the multi-model median anomaly, and the whiskers show the maximum model spread. Melt rates are calculated using the calculated $\gamma$ values and simulated temperature anomalies from Figure 10.

Translating melt rate anomalies into total ice shelf mass loss highlights the important role of ice shelf area in shaping
regional differences. A key caveat here is that we assume homogeneous basal melting across the ice shelf base, whereas in
reality, melting is typically concentrated near the grounding line. Unlike temperature anomalies (Figure 8), total ice shelf mass
loss patterns are influenced by both the sensitivity to thermal forcing and the spatial distribution of ice shelves. Under *antwater*,
ice shelf mass loss varies from -40 Gt yr$^{-1}$ in the Bellingshausen region to +80 Gt yr$^{-1}$ in the Weddell region, while SSP5-8.5
simulations indicate a range from +55 Gt yr$^{-1}$ in Dronning Maud Land to +690 Gt yr$^{-1}$ in the Weddell. This pattern aligns
with observations (Figure 10), where high melt rates in the Totten region contribute less to total ice shelf mass loss due to its
relatively small ice shelf area, while the Weddell region experiences the highest mass loss, reflecting its extensive ice shelf
coverage.

The multimodel median and spread of these total ice shelf mass loss anomalies in all IMBIE regions are summarized in
Figure 12, providing a pan-Antarctic perspective on regional disparities. This figure demonstrates the accumulated uncertainty
and the large regional differences in the loss of mass on the Antarctic ice shelf due to global warming (SSP5-8.5) and meltwater
(*antwater*) feedbacks. These differences are shaped by a combination of regionally varying ocean temperature anomalies, ice
shelf area, and variations in melt sensitivity, highlighting the intricate and uncertain nature of Antarctica's future mass balance.





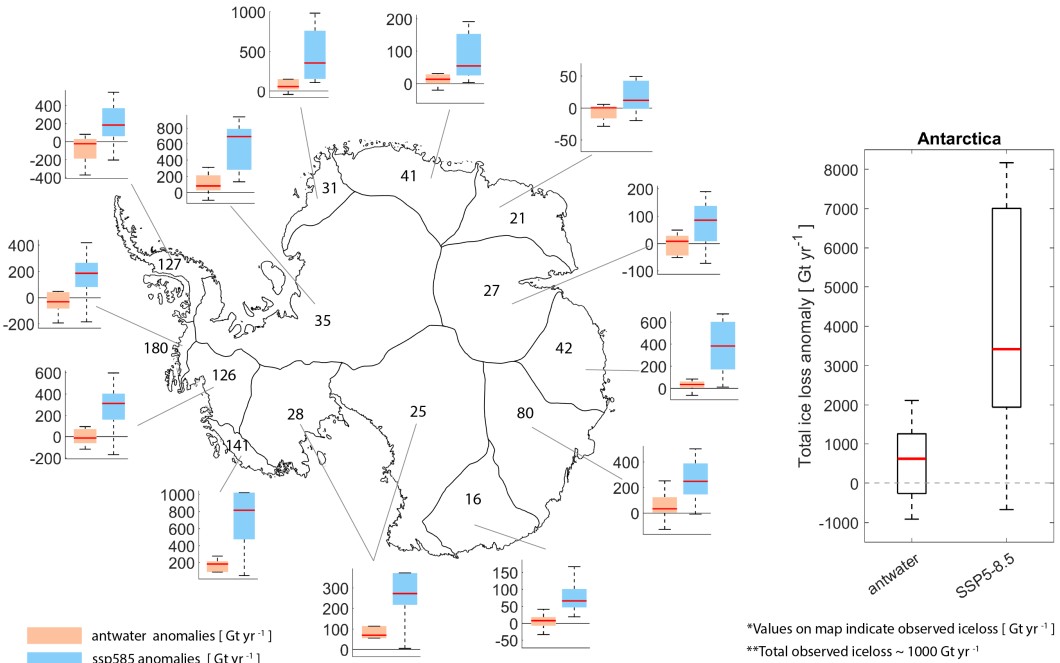

**Figure 12.** Map of Antarctic regions with box plots of multi-model anomalies of total ice shelf mass loss (melt rates multiplied by total ice shelf area) in response to the *antwater* experiments (orange) and under SSP5-8.5 (blue). On the right, a box plot shows the sum of all regions, representing the total uncertainty in ice shelf mass loss along the entire Antarctic continent due to the oceanic basal melting feedback in the meltwater experiments (*antwater - piControl*, comparing the last 10 years) and under the SSP5-8.5 simulations (comparing 2090–2100 with 2015–2025).

## 4 Discussion

### 4.1 Does an Asymmetric Temperature Response Lead to a More Symmetric Ice Shelf Mass Loss Around the
Continent?

One of the most striking findings of this study is the projected shift in spatial ice shelf mass loss patterns. Present-day observations indicate that West Antarctica is the primary contributor to basal melt-driven ice shelf mass loss (Figure 13a), and recent modeling studies predict that a future increase in West Antarctic ice shelf melting is unavoidable (Naughten et al., 2023). However, our results suggest that future ice shelf mass loss could become more evenly distributed around the continent. This

reflects the emerging "cold shelf to warm shelf" paradigm, where historically colder and more stable ice shelves—primarily in East Antarctica—begin to contribute more significantly to total mass loss, thereby redistributing melt around the continent on an elevated baseline. This also aligns with emerging evidence suggesting that East Antarctica may be more vulnerable to melting than previously assumed (Herraiz-Borreguero and Naveira Garabato, 2022). Although both the simulations *antwater* and SSP5-8.5 show widespread subsurface ocean warming, this warming is modulated by a dampening meltwater or feedback

in the Bellingshausen under *antwater*, leading to an asymmetric temperature response on the continental shelf (Figure 7a). As a result, East Antarctica is projected to play an increasingly significant role in ice shelf mass loss. This is illustrated in Figure 13b, where the multi-model median ice shelf mass loss anomalies from the *antwater* experiment are added to present-day



shelf mass loss trends. The resulting pattern reveals an increasing relative contribution from East Antarctica, primarily due to cooling anomalies that suppress basal melting and reduce ice shelf mass loss in parts of West Antarctica.

Currently, the highest ice shelf mass loss occurs in the Bellingshausen region. However, when *antwater* anomalies are incorporated, the F-G sector (between the Amundsen and Ross Seas) emerges as the largest contributor, with the Totten region becoming equally significant to the Bellingshausen and Amundsen sectors. The Weddell region also gains prominence, while the Ap-B, Eastern Ross, and Amery sectors remain the smallest contributors. In particular, the increasing importance of the F-G region is striking, as it shares a similar warm continental shelf with the Bellingshausen and Amundsen regions in today's
climatology, but remains unaffected by the cooling anomalies that develop in those regions under *antwater*.

     Although smaller in magnitude, an asymmetry in temperature anomalies is also evident in SSP5-8.5, at least in some models (Figure 7b), with a weaker warming signal in the west compared to the east. Similarly, when the median SSP5-8.5 ice shelf mass loss anomalies are added to the present-day mass loss, East Antarctica and the Weddell region exhibit ice shelf mass loss rates comparable to the Amundsen and Bellingshausen sectors. In particular, the F-G sector once again emerges as the
largest contributor to total mass loss by basal melting. In contrast, the Eastern Ross, Ap-B, and Amery regions continue to show relatively modest contributions.

     We note, however, that our *antwater* multi-model median is heavily influenced by models showing a strong negative feedback in response to meltwater in West Antarctica, and our results are thus very model-dependent. For example, some models—such as AWI-ESM—show no negative temperature anomalies anywhere along the coast in the *antwater* simulation. The cause of
these differences remain unclear, but notably, AWI-ESM is the only model in the ensemble that employs an unstructured mesh, which may contribute to these differences. It is also the only model that does not exhibit a "warm shelf" regime in the Bellingshausen region, where other models tend to show pronounced negative anomalies (Figure 9).

     Model dependency is further exemplified by comparing our results with those of Naughten et al. (2023). They use a regional ocean–ice model forced by CESM1 climate forcing and project a continental shelf warming of 1.39 °C in the Amundsen Sea
region under the RCP8.5 scenario. This is broadly consistent with our results: our third quartile spans 0.5 to 1.0 °C under SSP5-8.5, with CESM2–the successor of CESM1– simulating a warming of 0.55 °C in this region 8b), which is lower than the CESM1 projection under RCP8.5. Compared to other models, CESM1 thus provides a high-end estimate of shelf warming in this region. The model used by Naughten et al. (2023) includes ice-shelf basal melt and the associated heat and freshwater fluxes, but is not coupled to an ice sheet and therefore does not account for additional freshwater input from ice sheet runoff. As
such, it cannot capture the feedback mechanism investigated in our *antwater* experiment. Even if this feedback were included, CESM2 does not simulate a negative feedback in this region under *antwater*, suggesting that CESM1 may not either. Hence, as also noted by Naughten et al. (2023), it remains uncertain whether CESM1 overestimates or underestimates future freshening and, in turn, warming. This does not suggest that continued warming in the Amundsen Sea is unlikely. However, given that several models in our ensemble project a potential negative feedback in response to additional meltwater, and considering
that CESM1 produces stronger regional warming than most models, it is possible that future warming in this region may be somewhat lower than projected by Naughten et al. (2023).

     In addition to intermodel differences, other processes may also affect the regional patterns. For example, in regions where meltwater induces a cooling response, including ice shelf feedbacks would likely reduce melt rates, thereby weakening the freshening that initially drives the cooling—highlighting the complex and nonlinear nature of ocean–ice shelf interactions.
Moreover, climate-change-driven shifts in other freshwater sources not included in the *antwater* experiments—such as increased iceberg calving, altered sea ice melt and growth patterns, and regional variations in ice sheet surface mass balance (e.g., precipitation minus evaporation)—may significantly influence the spatial distribution of freshwater (as shown by Lockwood et al. (2021) and Goddard et al. (2017)) and, in turn, modulate regional ocean warming or cooling.

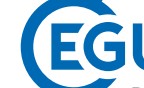

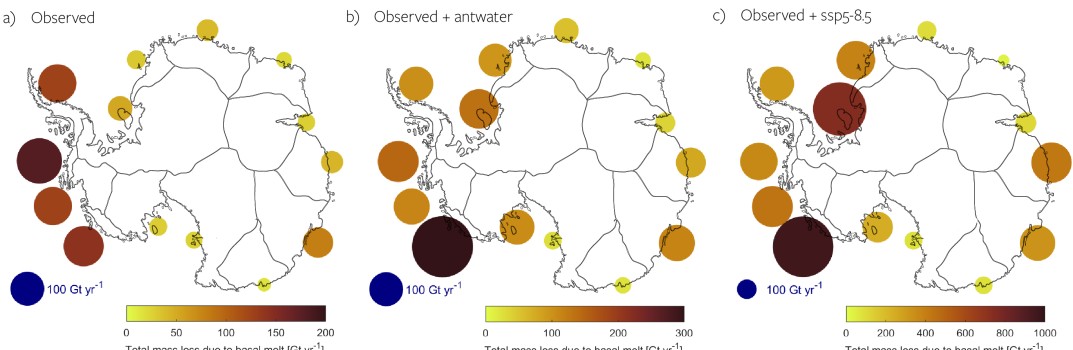

**Figure 13.** Map of Antarctic regions showing regional differences in total ice shelf mass loss due to basal melting: a) observed, b) observed plus multi-model median anomalies from meltwater perturbation experiments (*antwater - piControl*, comparing the last 10 years), and c) observed plus multi-model median anomalies from the SSP5-8.5 scenario (comparing 2090–2100 with 2015–2025).

### 4.2 Does Meltwater Discharge Amplify or Suppress Global Warming-Induced Continental Shelf Warming?

Anomalies of ice shelf basal melting due to ocean warming under the SSP5-8.5 scenario show high uncertainty (Figure 12a). Integrated across Antarctica (Figure 12b), the multi-model median indicates a loss of +3300 Gt yr$^{-1}$ by the end of the century, yet the range spans from -800 Gt yr$^{-1}$ to +7900 Gt yr$^{-1}$. The large spread reflects the large variability in ocean warming projections and their implications for basal melt and ice shelf mass loss. Remarkably, some models still project a net reduction in ice shelf mass loss (negative anomalies) even under the strong SSP5-8.5 warming scenario.

Our analysis shows that the feedback from the additional meltwater input is of similar magnitude, which may reinforce or compensate the global warming signal in different regions. Integrated across the continent, the multi-model median response to *antwater* simulations indicates a total additional ice shelf mass loss due to basal melt of approximately +750 Gt yr$^{-1}$ under preindustrial control conditions. However, there is considerable variability between models, with some projecting a reduction in total ice shelf mass loss (negative anomalies), while others estimate increases of up to +1900 Gt yr$^{-1}$. For context, the
current observed ice shelf mass loss due to basal melting is approximately 1000 Gt yr$^{-1}$ (Rignot et al., 2019; Paolo et al., 2023; Davison et al., 2023).

    Given that the *antwater* and SSP5-8.5 anomalies are driven by different forcings, their responses are not linear, and their effects cannot be assumed to be simply additive. The full uncertainty and combined impact of Antarctic meltwater discharge and global warming under SSP5-8.5 can only be properly assessed through scenario simulations that include additional meltwater
forcing—such as those proposed in the Tier 2 simulations of the SOFIA experiment (Swart et al., 2023)—and will therefore be the subject of future studies.

    It remains uncertain whether Antarctic meltwater discharge will amplify or suppress global warming–induced shelf warming. However, our results suggest that if the feedbacks observed in the *antwater* simulations also occur under scenario forcing, then warming may be dampened in regions exhibiting a negative feedback and further intensified where a positive feedback
dominates. Given that the feedbacks and warming signals are of similar magnitude, these interactions are likely to be significant—highlighting their potential impact on future changes in Antarctic ice shelf dynamics.




### 4.3 Is Cooling in West Antarctica Driven by Regional Ocean Connectivity?

Although the meltwater perturbations in the *antwater* experiment are applied uniformly around the Antarctic continent, some
models produce markedly different patterns of temperature anomalies along the shelf, while others show more consistent re-
sponses, raising the question of whether certain underlying mechanisms are active in some models but absent in others. In
particular, the cooling anomaly in west Antarctica is evident in 6 of 10 models, but not all, underscoring considerable un-
certainty in how the ice shelves in this region will respond to increased freshwater input. Using a high-resolution different
configuration of the ACCESS model (ACCESS-OM2-01), Moorman et al. (2020) demonstrated that meltwater-induced fresh-
ening strengthens along-slope and along-shelf density gradients, accelerating the westward-flowing ASC and ACoC. They
propose that this enhanced circulation advects cold waters from the Weddell Sea into the warmer Bellingshausen and Amund-
sen Seas, leading to shelf cooling in West Antarctica (Moorman et al., 2020). Beadling et al. (2022) and Tesdal et al. (2023)
proposed that the same mechanism could be at play in their respective meltwater perturbation experiments. The connective
pathway from the Weddell Sea around the Antarctic Peninsula was first identified by Heywood et al. (2004), has recently been
mapped with Lagrangian particles by Dawson et al. (2023), and is further supported by seal-mounted CTD data showing that
the route continues into the Bellingshausen Sea (Schubert et al., 2021).

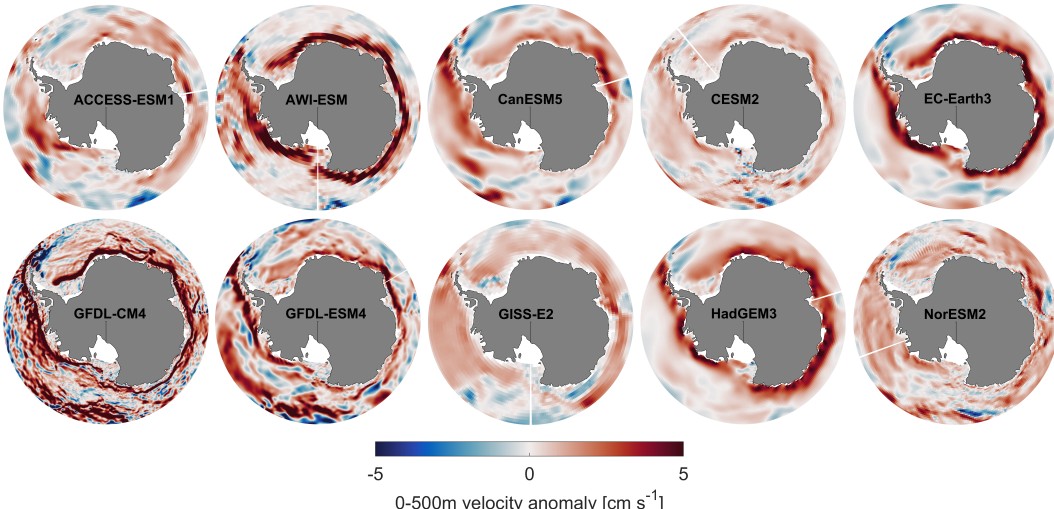

**Figure 14.** Spatial anomalies of vertically averaged (0-500 m) along-slope velocity (cm s$^{-1}$, with positive values indicating westward flow)
in response to the meltwater perturbation experiments (*antwater - piControl*, comparing the last 10 years). Red values indicate a strengthening
of westward currents (e.g., the along-slope Antarctic Slope Current (ASC) and on-shelf Antarctic Coastal Current (ACoC)), while blue values
indicate a weakening of westward currents or a strengthening of eastward currents. 0-500 m is chosen as this is the vertical extent of the
ASC.

Our results support this interpretation, showing that, in our multi-model ensemble, strong cooling along the West Antarc-
tic shelf generally coincides with Antarctic Slope Current (ASC) intensification. Figure 14 presents the spatial anomaly of
vertically averaged along-slope velocity in the upper 500 m, revealing a pronounced strengthening of the westward flowing
ASC (red shading) near the shelf break in all models. Models with the strongest ASC enhancement (AWI-ESM, EC-Earth3,
GFDL-CM4, GFDL-ESM4, HadGEM3) have more freshwater trapped in the boundary (Figure 5 and 6) and also exhibit the
most pronounced cold anomalies, reinforcing the link between ASC dynamics and shelf temperature changes. This could also





represent a positive feedback, where a strong mean-state ASC traps more freshwater near the coast, further strengthening the ASC.

However, ASC strengthening alone does not fully explain the presence or absence of coastal cooling. For example, some models that do not show a cooling anomaly in the Bellingshausen or Amundsen regions—such as AWI-ESM, CanESM5, CESM2, and NorESM2 (Figure 4a), also display a strengthened ASC. This discrepancy may arise from differences in the mean state. For instance, Figure 9 shows that AWI-ESM does not feature a warm regime on the West Antarctic shelf to begin with, instead exhibiting a substantial cold bias. Additionally, differences in how models simulate the response to *antwater* in the Weddell Sea could influence the advection of anomalies, further affecting regional responses.

In both Moorman et al. (2020) and the combined studies of Beadling et al. (2022) and Tesdal et al. (2023), the development of cold anomalies along the western Antarctic shelf coincides with a reduction in DSW production in the Weddell Sea. This shift involves a transition from dense shelf water formation to the production of lighter dense waters that are too buoyant to overflow from the continental shelf and cascade to the ocean bottom. As a result, some of the cold anomalies in West Antarctica may be linked to the westward advection of cold, fresh dense waters from the Weddell Sea around the Antarctic Peninsula into the Bellingshausen and Amundsen Sea sectors. The assessment of ocean age in Beadling et al. (2022); Moorman et al. (2020); Tesdal et al. (2023) supports this mechanism, as recently ventilated waters emerge in West Antarctica. Furthermore, Morrison et al. (2023a) highlighted the connection between the export of the Weddell Sea DSW and the variability of the temperature in the West Antarctic in high resolution ocean model simulations.

If the cooling observed along the west Antarctic shelf is indeed driven by this suppression of DSW formation and the subsequent westward advection of cold waters around the Antarctic Peninsula, Beadling et al. (2022); Moorman et al. (2020); Tesdal et al. (2023) suggest that a negative bottom age anomaly would be evident in the region (as older CDW is being replaced by recently ventilated Weddell Sea waters). Although not all models include seawater age as an output variable, for those that do, we analyze the bottom age anomaly on the continental shelf in the *antwater* simulation, as shown in Figure 15. GFDL-CM4, GFDL-ESM4, and HadGEM3 exhibit pronounced negative bottom age anomalies in West Antarctica, coinciding with strong and consistent cold temperature anomalies in this region. ACCESS-ESM1 and CESM2 also display a negative bottom age anomaly, albeit much weaker, which aligns with the somewhat weaker and cold anomaly/small warm anomaly, respectively. In contrast, NorESM2 and CanESM5 reveal positive bottom age anomalies in West Antarctica, consistent with their warm temperature anomalies. These findings support the hypothesis that the observed cold anomalies along the west Antarctic shelf may be driven by advected anomalies and upstream processes originating in the Weddell Sea. However, fully confirming this mechanism requires a detailed investigation of the water mass transformation processes and DSW production, which will be the focus of a future study.

### 4.4 A combination of uncertainties from multiple factors

In the Introduction, we identified multiple factors contributing to uncertainty in simulating basal melting feedbacks: (a) model-dependent climate responses and feedbacks to meltwater discharge, (b) the complex interactions between the open ocean and the continental shelf, (c) uncertainties in basal melt parameterizations due to limited observations, (d) the interplay between meltwater-induced and global warming-driven changes, and (e) biases in climate models' representation of Southern Ocean water masses and near-shelf dynamics.

Our results show that the model spread in total ice shelf mass loss under *antwater* and SSP5-8.5 reflects the combined effect of all these uncertainties. This highlights the challenge of reliably downscaling and projecting future coastal ocean conditions that will influence Antarctic ice shelf mass loss. While we cannot isolate the dominant source of uncertainty, we can provide insights into each contributing factor and offer recommendations for future research:





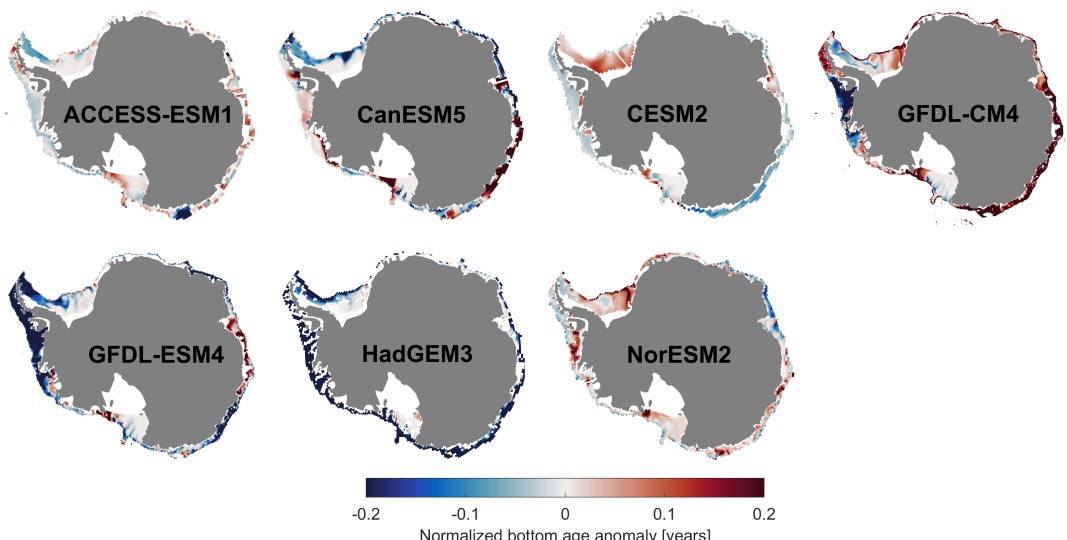

**Figure 15.** Spatial bottom age anomaly (years since surface contact) along the Antarctic shelf (southward of the 1000 m isobath) for the last 10 years of the *antwater* experiment. To normalize, the mean bottom age at each grid cell is divided by the spatial mean bottom age in the Southern Ocean (south of 30°S and deeper than 4000 m) from the *piControl* simulations. Negative values mean older water is being replaced by more recently ventilated water, and positive values indicate less ventilation of bottom waters. Adapted from Beadling et al. (2022), Figure A3. The variable "agessc" was not available for the AWI-ESM1, EC-Earth3, and GISS-E2 models.

a) **Multi-Model Perspectives on Meltwater Response**

The SOFIA multi-model ensemble provides, for the first time, a direct comparison of how different climate models respond to identical meltwater forcing. Our results reveal consistent large-scale responses, including deep-ocean warming and shelf warming in some regions while others experience cooling. However, substantial differences exist in both magnitude and regional distribution. This finding reinforces the importance of using multi-model ensembles rather than relying on single-model interpretations, similar to other climate model studies. Other alternatives to large model ensembles are to evaluate models against observations and select those that best represent Southern Ocean conditions (e.g., Barthel et al., 2020), or to avoid models that are outliers in their response to either meltwater or global warming. The intermodel spread in temperature anomalies remains a significant source of uncertainty in projected ice shelf mass loss.

b) **Regionality of Meltwater Feedbacks**

First and foremost, our results show that meltwater and global warming responses are highly regionally dependent. Zonal averages can be misleading and are inappropriate when considering changes along the different continental shelf regimes (Thompson et al., 2018). By analyzing regional responses, we reduce the uncertainties inherent in studies using zonal averages. For example, previous studies have suggested that the basal melting feedback to meltwater is predominantly positive (e.g., Bronselaer et al., 2018), which is accurate for the open-ocean Southern Ocean as a whole but not for individual shelf regimes (Figure 5 and 6). Our findings also emphasize that local continental shelf dynamics—including slope front changes, deep shelf water (DSW) production, and strengthening of the Antarctic Slope Current (ASC) or Antarctic Coastal Current (ACoC)—are key to understanding meltwater impacts. Simply extrapolating offshore ocean properties onto the continental shelf is, therefore, inadequate. Instead, regional connectivity and advective pathways (Dawson et al., 2023) must be considered. We thus recommend that future studies avoid zonal mean approaches and instead focus on



regional responses on the continental shelf when assessing ice-ocean interactions. Additionally, similar to findings by Tesdal et al. (2023) and Beadling et al. (2022), we find that coarse-resolution climate models (all in this study, except GFDL-CM4) poorly represent shelf break dynamics, likely influencing their simulated responses. This underscores the critical role of model resolution in capturing key feedbacks and highlights the importance of assessing whether models adequately resolve features such as the ASC and DSW formation, both of which strongly affect freshwater retention and the strength of ocean–ice feedback mechanisms.

c) **Uncertainty in Basal Melt Parameterization**

Despite recent advances, basal melt parameterizations (Jourdain et al., 2020) remain a key source of uncertainty. Recent studies suggest that a quadratic relationship provides a reasonable approximation (Burgard et al., 2022), but a major challenge lies in the availability of unbiased observational data. For instance, overestimated temperatures in Dronning Maud Land and underestimated temperatures in the Amundsen Sea (Section 3.3) influence the inferred melt rate coefficient $\gamma$, leading to uncertainties in mass loss estimates. Nevertheless, the updated climatology used in this study incorporates the most comprehensive observational datasets to date, including marine mammal-derived measurements from beneath ice shelves. As a result, this climatology likely introduces fewer biases than global ocean products, which are known to have significant errors (Jourdain et al., 2020). A key takeaway from our work is the importance of regionally calibrated melt rate parameterizations rather than assuming a uniform $\gamma$ across Antarctica, supporting findings from Jourdain et al. (2020) and Lambert et al. (2024). Additionally, as observational ocean climatologies are updated, they should be paired with updated basal melt rate datasets (e.g., Paolo et al., 2023) to ensure consistency.

d) **The Interaction Between Meltwater and Global Warming**

Understanding the combined effects of meltwater discharge and global warming remains a challenge. Our comparison of *antwater* and SSP5-8.5 simulations provides insights into the consistency (or lack thereof) in their respective impacts on basal melting. However, caution is needed when interpreting these comparisons. While SSP5-8.5 does not explicitly include meltwater forcing, the two experiments cannot simply be summed due to potential nonlinearities and missing feedbacks. Nevertheless, comparing these scenarios offers insight into whether their effects are likely to amplify or counteract each other. Future experiments, such as the Tier 2 SOFIA simulations, which include both greenhouse gas forcing and additional meltwater (Swart et al., 2023), will provide a clearer picture of the combined effects. One key takeaway from our work — relevant for future Tier 1 vs. Tier 2 comparisons — is that the SSP5-8.5 simulations already include additional freshwater input through enhanced hydrological cycle and increased P-E. This freshwater input, which is comparable in magnitude to the 0.1 Sv forcing used in *antwater*, suggests that similar mechanisms may be active in both experiments, albeit masked or counteracted by competing warming trends.

e) **The Role of Model Bias**

Model biases remain a major source of uncertainty in projecting future Antarctic mass loss (Barthel et al., 2020). Our results confirm that climate models struggle to accurately represent Southern Ocean water masses, with biases in the mean state also affecting responses to warming and meltwater perturbations. Additionally, biases in *piControl* climate simulations are often larger than the temperature anomalies driven by meltwater and nearly as large as those projected under high-emissions scenarios by the end of the century (Figures 3b, 3c, and 8c). These findings underscore the need for a multi-model perspective in future studies. Improving the representation of Southern Ocean mean-state properties should also be a high priority for the climate modeling community. Antarctic water mass biases are particularly relevant when using global climate models to force regional simulations. We highly recommend evaluating models against observational datasets wherever possible and using caution when extrapolating real-world behavior in regions where model biases exceed the magnitude of the forced climate change signal. At the same time, it is important to acknowledge uncer-



tainties in the observational datasets themselves, the challenges of comparing different time periods (e.g., pre-industrial simulations with present-day observations), and the influence of internal variability, which may be undersampled in both models and observations.

## 4.5 Experimental caveats

While the SOFIA *antwater* experiment provides a valuable framework to assess the impacts of meltwater in the Southern Ocean, several simplifications should be considered when interpreting the results. A key limitation is that meltwater is evenly distributed across all grid cells adjacent to the Antarctic coast at the surface, whereas in reality, meltwater enters the ocean through a combination of basal melting, where freshwater is introduced at depth, and calving, where icebergs melt at the surface but often far from the coastline. Moreover, the input of Antarctic meltwater is not uniform between regions, with the highest contributions originating from the Amundsen and Bellingshausen Seas, where the ice shelf melt rates are most pronounced (Seroussi et al., 2020). The impact of the horizontal and vertical distribution of meltwater has been examined in individual modeling studies (e.g., Merino et al., 2018; Mackie et al., 2020a; Thomas et al., 2023), as has the inclusion of latent heat associated with freshwater perturbations (Thomas et al., 2023), highlighting their critical role in shaping regional ocean responses. For example, using the SOFIA simulations, Kaufman et al. (2025), show that Southern Ocean SST trends are more sensitive to freshwater fluxes concentrated along the Antarctic margin versus more spatially distributed fluxes. The Tier 3 SOFIA experiments (Swart et al., 2023) will explicitly test the influence of the spatial distribution of the meltwater and the latent heat effects, providing a more refined understanding of these processes. Another important consideration is that the experiment accounts only for the freshening effect of meltwater, while neglecting the heat loss associated with adding water at the local freezing point and the latent heat loss due to melting. Previous studies (e.g., Hattermann and Levermann, 2010) suggest that this omission could lead to an underestimate of ocean cooling feedbacks. Finally, while biases in modeled ocean properties have been discussed, climate drift in long simulations is not explicitly accounted for, and natural internal variability within the models may also influence the results. For instance, it is possible that specific events—such as episodes of deep convection or variability in the tropical Pacific—may be subsampled, potentially contributing to the signals observed, as suggested by Purich and England (2021).

## 5 Conclusions

Using the SOFIA Tier 1 ensemble, we examined the ocean's response to a 0.1 Sv meltwater perturbation (*antwater*) and the resulting feedback on basal melting. Although meltwater feedback is generally thought to amplify basal melting, our results show that its impact varies regionally, enhancing basal melting in some sectors while suppressing basal melting in others. The multimodel median reveals a warming feedback on the continental shelf in most regions; however, West Antarctica exhibits cooling or subdued warming in most models, particularly in the Bellingshausen and Amundsen Seas, and some models also indicate cooling in the Amery region. This highlights the limitations of zonal averages, which can obscure crucial regional differences across distinct continental shelf regimes, and aligns well with recent studies highlighting the importance of regional hydrography (Song et al., 2025).

The SOFIA simulations support existing hypotheses linking these asymmetric temperature responses to strong regional connectivity. Local shelf break dynamics, including a strengthened Antarctic Slope Front (ASF), limiting Circumpolar Deep Water (CDW) intrusion, an accelerated Antarctic Slope Current (ASC) advecting anomalous Weddell waters, and reduced Dense Shelf Water (DSW) formation, play a crucial role in shaping regional responses. These processes are poorly resolved in coarse-resolution models, which likely affects their ability to simulate realistic meltwater-induced changes. This further



implies that extrapolating offshore values onto the shelf is unrealistic, emphasizing the need for more refined approaches when assessing ice-ocean interactions.

A key question is how the effects of additional meltwater forcing compare to, and are modulated by, broader global warming–induced trends. While the SOFIA Tier 2 scenario experiments (Swart et al., 2023) — including added meltwater forcing under future climate conditions—are designed to address this, they are not yet available. In the meantime, by comparing the *antwater* experiments with SSP5-8.5 simulations—which reflect a wider range of global warming impacts, including substantial freshwater input from enhanced precipitation — we find that SSP5-8.5 generally produces a more spatially uniform warming along the Antarctic margin. On average, continental shelf warming is approximately three times stronger under SSP5-8.5 than under *antwater*. However, some models still exhibit cooling or reduced warming west of the Antarctic Peninsula, suggesting that similar mechanisms operate in both experiments but are masked or offset by competing warming trends in SSP5-8.5.

To translate these ocean temperature anomalies into basal melting, we updated and regionally calibrated a quadratic basal melt parameterization using a new regional ocean climatology and the most recent satellite-derived basal melt rates. This approach yields regional anomalies in ice shelf basal mass loss of the ice shelf from both the *antwater* and SSP5-8.5 simulations.

Integrated across the continent, the multimodel median suggests that the 0.1 Sv *antwater* forcing leads to an additional 750 Gt yr$^{-1}$ of ice shelf mass loss, while SSP5-8.5 simulations project an increase of 3,400 Gt yr$^{-1}$ by the end of the century. For context, current observational estimates place ice shelf mass loss at approximately 1000 Gt yr$^{-1}$, indicating a potential 75% increase driven by meltwater feedback alone and a 340% increase under high greenhouse gas emissions.

However, substantial uncertainty persists, as the model spread represents a convolution of multiple sources of error, including biases in the model's mean state, limitations in the experimental design, poorly constrained melt parameterizations, and the complex interactions between the open ocean and diverse continental shelf regimes, which may not be adequately resolved in coarse-resolution climate models.

Despite these uncertainties, one consistent key finding emerges: in the West Antarctic regions where the greatest ice shelf mass loss has been observed in recent decades, most models project either cooling or reduced warming in the *antwater* simulation, indicating a dampening feedback due to meltwater input. This suggests that East Antarctica may become equally important in terms of future ice shelf basal mass loss, consistent with a broader shift from "cold shelf to warm shelf". We therefore propose that the asymmetric temperature response to meltwater could drive a shift toward a more symmetric distribution of ice shelf mass loss across the continent — but at a higher overall rate. Future studies should assess whether reduced melt rates in West Antarctica could substantially lower the risk of marine ice sheet instability, which remains a key driver of high-end sea level rise projections from the West Antarctic Ice Sheet.

*Acknowledgements.* This study has received funding from the European Union's Horizon 2020 research and innovation programme under grant agreement No 101003826 via project CRiceS (Climate Relevant interactions and feedbacks: the key role of sea ice and Snow in the polar and global climate system). TH was supported by Research Council of Norway grant nos. 314570. The computations were performed on resources provided by Sigma2 - the National Infrastructure for High-Performance Computing and Data Storage in Norway, grant nos. NS9081K and NN9824K and nn11029k. RLB was supported under NSF Division of Polar Programs Grant NSF2319828. AGP, IJS, and MT were funded by the Deep South National Science Challenge (MBIE contract number C01X1412). AGP, IJS, and MT acknowledge additional support from the Antarctic Science Platform (University of Otago subcontract 19424 from VUW's ASP Project 4 contract with Antarctica New Zealand through MBIE SSIF Programmes Investment contract number ANTA1801). AGP, IJS, and MT wish to acknowledge the use of New Zealand eScience Infrastructure (NeSI) high performance computing facilities and consulting support as part of this research: New Zealand's national facilities are provided by NeSI and funded jointly by NeSI's collaborator institutions and through the Ministry of Business, Innovation & Employment's Research Infrastructure programme. MHE was supported by the Australian Research Council grant



nos. SR200100008, DP190100494 and DP250100759. AP was supported by the Australian Research Council Special Research Initiative for Securing Antarctica's Environmental Future (SR200100005). IT was supported by the Italian National Recovery and Resilience Plan project TeRABIT (IR0000022—PNRR Mission 4, Component 2, Investment 3.1 CUP I53C21000370006) in the frame of the European

805    Union—NextGenerationEU funding. This research has also been funded by Ocean Cryosphere Exchanges in ANtarctica: Impacts on Climate and the Earth system, OCEAN ICE, Grant agreement ID: 101060452, 10.3030/101060452 Internal contribution Nr. 31.

*Data availability.* The data from the *antwater* experiments are provided by the SOFIA initiative and are available at: https://sofiamip.github.io/data-access.html (Swart et al., 2023). The *piControl* and SSP5-8.5 simulations are available through the CMIP6 archive (Eyring et al., 2016) via the Earth System Grid Federation (ESGF). For the analysis presented here, we accessed data through the Geophysical Fluid

810    Dynamics Laboratory (GFDL) node: https://esgdata.gfdl.noaa.gov/search/cmip6-gfdl/, and the Lawrence Livermore National Laboratory (LLNL) node: https://esgf-node.llnl.gov/projects/cmip6/. We kindly request that users cite the experimental design paper by Swart et al. (2023) when using SOFIA data in publications. Additionally, we recommend contacting the SOFIA community via https://sofiamip.github.io/ prior to data use, as updates or additional data may be available.

*Author contributions.* Conceptualization: MM, TH; Data curation: MM, RLB, NCW, AN, CG, SZ, CD, AJ, QL, AGP, AP, ZS, MT; Formal
815    analysis: MM; Writing – original draft: MM, TH, RLB; Visualization: MM; Writing – review & editing: All authors.

*Competing interests.* The authors declare that they have no conflict of interest.



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
