# Peer review of "Large Regional Differences in Antarctic Ice Shelf Mass Loss from Southern Ocean Warming and Meltwater Feedbacks"

_EGUsphere, 2025_

## Referee Comment (RC2)

**Review of Muilwijk et al. (for EGUSphere)**

**Overall Assessment**

This manuscript provides an insightful multi-model analysis of Southern Ocean regional responses to Antarctic Meltwater fluxes, including an estimate of the feedback on ice shelf basal melt. The key robust result is that Antarctic meltwater causes less-positive temperature anomalies along the continental shelf of the West Antarctic than for East Antarctica; since West Antarctica is currently experiencing large melt rates, they conclude that this meltwater feedback would tend to make melt rates more uniform. Beyond this headline result, the results are more mixed, because in some models "less positive" means a net cooling response, whereas in others it means "reduced warming". A particularly perplexing result is how similar the GFDL models' response to Antarctic meltwater is – in sign, magnitude, and pattern – to the SSP5-8.5 response.

The manuscript is long but I think worth it, as I anticipate the various discussions of model-specific behaviors (complemented by extensive referencing to relevant literature) will inspire future work to resolve the open questions about which model responses are correct. The authors have made a convincing case that the value of this controlled model intercomparison project outweighs the caveats about the unrealistic distribution of meltwater. (Additionally, they point out that several of the most important caveats will be addressed by later phases of this collaborative project.)

I recommend the manuscript be accepted for publication after addressing the following comments.

**Main comments**

**Needs more precise language around West Antarctic cooling induced by meltwater.**

In various places, but most importantly in the concluding paragraph (L. 783-786) and the abstract (L. 10-11), it is stated that the "cooling or reduced warming" on the West Antarctic continental shelf "suggest[s] a negative feedback" or "indicated a dampening feedback". This is misleading, since more than half of the models in the ensemble still show net warming on the continental shelf, meaning a net positive/amplifying feedback. I believe you are trying to make one or both of the two subtler points:

- 1) The net feedback in West Antarctica is less positive than the circumpolar average, suggesting compensation by a negative feedback process.
- 2) The two models that you seem to have the most confidence in (GFDL ESM4 and CM4), because of their relatively small biases and higher resolution, exhibit a clear net negative feedback on West Antarctic temperatures in response to meltwater forcing (which, surprisingly, arises even under SSP5-8.5 forcings), suggesting that the feedback in the natural world is net negative.

**Figure 3** - How is P-E + Runoff larger than the total in some models, e.g. HadGEM3, under SSP5-8.5 forcing? This implies the residual (sea ice melt/freeze and iceberg calving) is negative. But surely sea ice volume does not increase from 60°S-90°S under such high forcing... This either needs to be explained here or else there needs to be a concrete statement about how this is resolved in Pauling et al. (in prep), and this companion manuscript needs to be made accessible to the reviewers. This result makes me also wonder what is going on in the other models...

**Minor comments**

- L. 114 Remove "both" since you then have 3 references after
- L. 172 Missing citation
- L. 757-759 I don't understand the flow of this sentence. Misplaced commas?

---

## Author Comment (AC1)

Dear Editor and Reviewers,

We sincerely thank you for the thorough and constructive reviews of our manuscript. We greatly appreciate the time and effort you dedicated to examining the details of our work. Your insightful comments have helped us substantially improve the clarity, focus, and overall quality of the paper.

Our detailed, point-by-point responses are provided below in blue font. If permitted, a tracked-changes version of the revised manuscript will be submitted for your reference. The main revisions reflect the major issues raised by the reviewers. In particular, we have carefully revised the manuscript to reduce unnecessary detail as noted by both reviewers, which has overall reduced the length of the paper.

On behalf of all co-authors,
Morven Muilwijk

**Reviewer #1**

In this study, the response of the continental shelf properties to Antarctic freshwater and global warming in a multimodel ensemble is investigated. The response shows a large regionality and this is further used to highlight that for ice loss estimates a regional melting rate should be applied rather than a uniform value. The conducted analysis in the manuscript is sound with the shown results, yet the manuscript is very long and contains a lot of details, which leads to the fact that some sections are difficult to follow. Further, some sections would need some more information for clarification. Overall, I am convinced that this manuscript will be suitable for publication after some revision. I recommend some careful text revision and some improvements on figures below.

**Major Comments:**

Overall, the manuscript is very long, and sometimes the authors explain parts very detailed and do not refer to them later on in the manuscript. E.g., lines 59 - 78 ("isolation" and "homogenization" - these terms are not used in the further manuscript); line 192 ("In addition, future…"  - this sentence does not contribute to the story line and could be removed); line 412 (" The Amery region…" this information could be interesting; yet it is not used further in the manuscript, so is it important to mention it here?),... I recommended going through the manuscript and revising if all mentioned information is needed to support the main findings of the manuscript. For me the main findings are: the regional changes of freshwater release as well as the importance of a regional melt rate than an Antarctic wide for estimating ice loss. If these are not the desired main aspects by the authors, please make them more clear.
We thank the reviewer for this helpful overarching comment. In response, we have carefully reviewed the full manuscript and removed or condensed several sections where the level of detail was not essential for supporting the main results. Specifically, we have shortened or eliminated descriptive passages that were not referenced later in the manuscript (including those noted by the reviewer) and tried to streamline overly detailed explanations. Our track-changes document will show exactly which sentences have been removed entirely. We believe these revisions improve the focus, readability, and coherence of the manuscript. We agree with the reviewer on the main findings and have attempted to make these more clear in the abstract as well.

Section 2.3.: The authors refer the reader to Zhou et al 2024a, for more details about climatology. Yet this work is "in prep" and therefore the information about how the data is treated/coverage is not available and makes it questionable. I would recommend some details and maybe plots like the reference field for Figure 5 and 6 in a supplementary material to provide more understanding on how the data is handled and combined. The paper describing the climatology (Zhou et al., 2024a) has since been submitted and will be available as a public preprint within a few days. This paper now provides full documentation of the data processing, coverage, and merging methods. We will update the manuscript

to cite the published preprint, ensuring that all relevant methodological details are now accessible to the reader. As these details are now publicly available, we did not add supplementary plots.

Section 2.4 and 2.5: In the current state, the structure is a bit hard to follow, as some parts have been explained in previous parts but other parts are coming later. I would recommend first to introduce how the regional melt coefficient is calculated using the data from 2.5 and then how the basal melt rates are estimated. Further, there is a misbalance between how data is introduced in Section 2.5, where the basal melting data set is described very detailed and even discussed, while the ice shelf draft is very brief. This makes me wonder, if all the details about the melting data set is needed. Regarding this, it is worth considering if all those details about the data set are a contribution, e.g. the purpose of lines 296-297 is not clear. Lastly, Table 2 does not exist. We are grateful for this observation. In the revised manuscript, we reorganized Section 2.4 to present the methodology in a clearer, more logical order as suggested by the reviewer.. We also streamlined Section 2.5 by reducing the detail of the basal melt dataset and removing the sentence previously discussing its comparison with earlier products. The section now introduces both the melt and draft datasets at a comparable level of detail. The reference to the Table has been removed.

Sections 4.1 - 4.3: I like the questions asked, yet the answers are not very clear in the respective text. It would be great if the authors could highlight a clear answer to their asked question. Section 4.3 adds a lot of extra analysis and yet the authors conclude with "However, fully confirming this mechanism requires a detailed investigation of the water mass transformation processes and DSW production, which will be the focus of a future study" (lines 649-651). I am wondering if all of this needs to be detailed in such a long manner.
We thank the reviewer for this helpful comment. In the revised manuscript, we now provide a clear, explicit answer at the beginning of each subsection (4.1–4.3) to address the questions posed in their titles. We also streamlined and shortened parts of the discussion to improve clarity and avoid repetition. Specifically Section 4.3 has been shortened by reducing repetition, condensing model-specific details, and shortening the mechanistic discussion. The overall length has been reduced and the flow improved in line with the reviewer's suggestion.

**Specific Comment:**

Line 78-92: I really like these sentences and they are very clear. Maybe it is worth having them as a stand alone paragraph.
Thank you, this is a very good suggestion. We made this a stand alone paragraph.

Line 86: It would be helpful if the resolution of the 2 models are mentioned.
Agreed. The resolution (0.25 and 0.5 degrees) have been added.

Line 138 - 163: I would recommend restructuring these paragraphs to be more reader friendly, the current structure requests the reader to go back and forth between the different paragraphs to understand how the goals, defined in lines 138-143, are tackled.
Thank you for this good suggestion. In the revised manuscript, we restructured these paragraphs so that each study objective (a–e) is now addressed in its own clearly marked paragraph. This makes it immediately clear how each goal is tackled without requiring the reader to move back and forth between paragraphs.

Line 180: It would be worth stating that these are the same models as in the antwater analysis.
Good suggestion. A small note has been added stating "from the same models".

Figure 2: This figure could describe the methodology better, e.g. annotate the arrows by saying which data is feeding into the regional meltwater rate. Thank for this suggestion. We agree and an improved

figure 2 will be included in the revised manuscript. This new figure now includes some more details on the various steps.

Line 195: "Precipitation - evaporation patterns …" - is this based on observations or models? This is based on models. We now clarified this in the text.

Line 197: "and some models reroute precipitation over the Antarctic continent directly into the ocean as runoff and/or calving" - this phrasing sparks the question what other models are doing. Very good point. We are actually not sure if there is an alternative approach. We can however, not be 100% sure if all models do this and have therefore rephrased to the following: "Furthermore, models without an interactive ice sheet likely reroute precipitation over the Antarctic continent directly into the ocean as runoff and/or calving. "

Figure 3a and respective lines: The decomposition of the total freshwater fluxes in the SSP5-8.5 scenario in P-E and runoff, shows that there must be a residual term. Which fluxes are represented by the residual? Are those calving and sea ice growth as mentioned in line 204? Please clarify. Correct. The residual fluxes are calving and sea ice growth/melt, which are sadly not available for all models. We have clarified this in the text.

line 231: Please describe the definitions from Barthel et al (2020) not everyone is familiar with them Thanks for suggestion this. The following clarification has been added: "which use a combination of bathymetry shallower than 1000 m and fixed northern boundary at 74°S."

line 311: What resolution is referred to as "coarse" here, as coarse is a subjective word, it is worth mentioning an order of magnitude. Agreed. we have added that we refer to models with approximately 1 degree horizontal as course.

Line 324-326: Is this general information or pointing to a particular figure? If this is general information, is it needed here as a separate paragraph, otherwise please reference the according figure. This is more general information and has now been included as introduction to the next paragraph rather than a separate paragraph.

Lines 351- 356: The discussion of the limitations of the climatology without showing the climatology is quite difficult. Please add the climatology, see comment on Section 2.3 above. As noted above, the paper describing the climatology is now available and has been updated in our citations. We now explicitly refer to this paper in Lines 351–356 and have clarified the text to make the discussion more transparent.

Line 363-365: "However, regional…" - the mentioned pattern is not easy to spot in the figures 5 and 6. For me it is only visible in 2 models, the GFDL models, and no common cooling signal in the SSP5-8. 5 simulations. Please provide more guidance where to see this information. Thank you for this comment. We agree that we might have overstated this pattern. We have rephrased to the following: "Some regional differences also appear in the SSP5-8.5 simulations, though they are much weaker than in the \textit{antwater} experiments, and only the GFDL models show any indication of cooling near the Antarctic Peninsula."

Line 375: "this regime"- it is not clear which regime is referred to. Good catch. We have added "warm" as this refers to one of the three shelf regimes defined by Thompson et al. (2018) introduced earlier on in the paper.

Line 405: "AWI-ESM…" - there are several models in the right column of Figure 7, which do not show a cooling signal, e.g.: ACCESS-ESM1, CESM2, EC-Earth3. Please revise this statement. We appreciate the reviewer for noticing that this phrasing was not entirely accurate. We here meant to refer to

"antwater" only and not the SSP5-8.5 simulations. We have rephrased this sentence to the following: "AWI-ESM is the only model that does not exhibit a cooling response to the \textit{antwater} forcing in any region around the continent."

Line 432-435: Based on Figure 8, the intermodel spread is even larger in Amery. Please rephrase. Agreed. Amery has been added to this sentence.

Line 442: "Some models…"- I would suggest mentioning these models in brackets in the text. Thank you for this good suggestion. Models have been listed in the revised version.

Lines 445 - 455: It is not obvious why some models are described here in detail why others are not. Further there are so many details in this paragraph, which makes it hard to follow. I would recommend revising this paragraph. Thank you for this suggestion. Reading back we agree there were too many details here. The paragraph has been heavily reduced, keeping only the most important points. It currently reads as follows: "Models that exhibit a cold on-shelf response to antwater generally show the anomaly extending throughout most of the water column, from the shelf break toward the continent. However, the vertical and horizontal structure of this cooling varies across the ensemble, and in several models the signal is weak, confined to shallow layers, or offset by subsurface warming. Under SSP5-8.5, all models show robust shelf warming, largely confined to the upper 500 m and consistent with surface-driven warming under climate change. However, the Bellingshausen shelf warms less than other regions (Figure 8), and in the GFDL models cooling persists locally. This suggests that the mechanisms contributing to the \textit{antwater} cooling response may also modulate the SSP5-8.5 shelf warming, albeit more weakly (see Section 4.3)."

Lines 456 - 458: The purpose of this paragraph is not obvious and needs to be better included with the remaining text. Agree. We have decided to remove this paragraph entirely.

Line 540-545: Here it is worth reminding the reader that the sectors are shown in Figure 1b. Very good point. Reference to sector definitions in Figure 1b has been included.

Line 565: Which feedback mechanism? Here we refer to the warming/cooling response on the shelf. We agree however that this was not clear and thank you for noticing this. The sentence has been revised to: "As such, it cannot capture the freshwater-induced continental shelf warming/cooling feedback investigated in our \textit{antwater} experiment."

Line 616-621: Why is the along slope velocity computed for the upper 500m and the temperature and salinity fields in Figures 5 and 6 in the upper 1000m? "trapped in the boundary" - what exactly is meant? on the shelf or in the ASC? "most pronounced cold anomalies" - AWI-ESM does not show pronounced cold anomalies in Figure 5 second column. Please revise this paragraph. Thank you for pointing out these inconsistencies. We now clarify that the along-slope velocity is averaged over the upper 500 m because this depth range corresponds to the core of the ASC in the models. In contrast, temperature and salinity anomalies can extend deeper on the shelf. We removed the phrase "trapped in the boundary". Here we meant to indicate that some models retain more freshwater on the shelf which also indicates a stronger slope front. However, in line with shortening this entire chapter as suggested in the reviewer's main comment, this detail has been removed entirely. We also revised the paragraph to avoid implying that the relationship between ASC strengthening and cold anomalies applies uniformly across all models, including AWI-ESM.

Figure 5, 6, 9, 13: The use of discrete color levels would improve readability of the values. This is a very good suggestion. All these colormaps have been replaced with discrete colors.

Figure 7: Maybe a weak horizontal grey line indicating the +-0.5°C would help to interpret the figure. Very good suggestion as well. A weak horizontal grey line has been added.

Figure 12: What is the reference for the multimodel anomalies? Further, there are 2 annotations marked with * and **, I assume that * refers to the numbers in the different basins, yet what does ** refer to?. Thank you for noticing this. The annotations technically belong in the caption and have been removed from the figure itself and now included in the caption. The reference for the multi-model anomaly is their piControl state.

Figure 13: Currently, it is quite hard to read the values and compare the circle size changing in between the subplots. We agree and thank you for this suggestion. An improved figure has been included, where we keep the reference circle size the same to make visual comparison easier.

**Technical corrections:**

Line 172: Reference not included
Corrected.

Line 185: "an observational climatology" instead of "the observational climatology"
Corrected.

Line 225: Reference not included
Corrected.

Line 277: m is missing the dot above
Corrected.

Line 360: Is Figure 4 here referenced or Figures 5 and 6 ? Please correct either the "s" or the number.
It is Figure 4 that is referenced here. The "s" has been removed.

Line 363: "fourth column" —> third column
Corrected.

Figure 8: Add labels a, b, c as the text refers to them.
Labels have been added.

Line 440: "second column" —> third column
Corrected.

Line 446: NorESM2-MM —> NorESM2
Corrected.

Caption Figure 10: Formatting of Tf
Corrected.

Line 466: Methods —> which section exactly?
This was section 2.3. Section number has been added.

Line 616: Only ASC, it is already introduced
Corrected.

---

## Author Comment (AC2)

Dear Editor and Reviewers,

We sincerely thank you for the thorough and constructive reviews of our manuscript. We greatly appreciate the time and effort you dedicated to examining the details of our work. Your insightful comments have helped us substantially improve the clarity, focus, and overall quality of the paper.

Our detailed, point-by-point responses are provided below in blue font. If permitted, a tracked-changes version of the revised manuscript will be submitted for your reference. The main revisions reflect the major issues raised by the reviewers. In particular, we have carefully revised the manuscript to reduce unnecessary detail as noted by both reviewers, which has overall reduced the length of the paper.

On behalf of all co-authors,
Morven Muilwijk

**Reviewer #2**

This manuscript provides an insightful multi-model analysis of Southern Ocean regional responses to Antarctic Meltwater fluxes, including an estimate of the feedback on ice shelf basal melt. The key robust result is that Antarctic meltwater causes less-positive temperature anomalies along the continental shelf of the West Antarctic than for East Antarctica; since West Antarctica is currently experiencing large melt rates, they conclude that this meltwater feedback would tend to make melt rates more uniform. Beyond this headline result, the results are more mixed, because in some models "less positive" means a net cooling response, whereas in others it means "reduced warming". A particularly perplexing result is how similar the GFDL models' response to Antarctic meltwater is – in sign, magnitude, and pattern – to the SSP5-8.5 response. The manuscript is long but I think worth it, as I anticipate the various discussions of model-specific behaviors (complemented by extensive referencing to relevant literature) will inspire future work to resolve the open questions about which model responses are correct. The authors have made a convincing case that the value of this controlled model intercomparison project outweighs the caveats about the unrealistic distribution of meltwater. (Additionally, they point out that several of the most important caveats will be addressed by later phases of this collaborative project.) I recommend the manuscript be accepted for publication after addressing the following comments.
We thank the reviewer for the thoughtful assessment of our work and appreciate the recognition of the value of the SOFIA intercomparison and the usefulness of the regional analyses. We have addressed all comments below and are grateful for the positive evaluation.

**Main comments**

Needs more precise language around West Antarctic cooling induced by meltwater. In various places, but most importantly in the concluding paragraph (L. 783-786) and the abstract (L. 10-11), it is stated that the "cooling or reduced warming" on the West Antarctic continental shelf "suggest[s] a negative feedback" or "indicated a dampening feedback". This is misleading, since more than half of the models in the ensemble still show net warming on the continental shelf, meaning a net positive/amplifying feedback. I believe you are trying to make one or both of the two subtler points:

1) The net feedback in West Antarctica is less positive than the circumpolar average, suggesting compensation by a negative feedback process.

2) The two models that you seem to have the most confidence in (GFDL ESM4 and CM4), because of their relatively small biases and higher resolution, exhibit a clear net negative feedback on West Antarctic temperatures in response to meltwater forcing (which, surprisingly, arises even under SSP5-8.5 forcings), suggesting that the feedback in the natural world is net negative.
We thank the reviewer for this important clarification. In general, point one is the primary point we are trying to make, which is supported by the fact that most models either show a net cooling or at least a

reduced warming in West Antarctica. We agree that our original phrasing could be interpreted as implying that the ensemble generally exhibits a net negative feedback in West Antarctica, whereas in reality only a subset of models shows actual cooling, and most others (except AWI-ESM) only show reduced net warming. To avoid ambiguity, we have revised the abstract and the conclusions to explicitly distinguish between "cooling" and "reduced warming," and clarify that only a subset of models exhibits a net cooling response. We believe these revisions improve the precision and interpretability of the manuscript.

Figure 3 - How is P-E + Runoff larger than the total in some models, e.g. HadGEM3, under SSP5-8.5 forcing? This implies the residual (sea ice melt/freeze and iceberg calving) is negative. But surely sea ice volume does not increase from 60ºS-90ºS under such high forcing… This either needs to be explained here or else there needs to be a concrete statement about how this is resolved in Pauling et al. (in prep), and this companion manuscript needs to be made accessible to the reviewers. This result makes me also wonder what is going on in the other models…

Thank you for this very good comment, this issue has puzzled us as well. We double-checked the CMIP6 output and, unfortunately, the freshwater flux terms associated with sea-ice growth and melt are not available for HadGEM (nor for most other models). This makes it impossible to close the freshwater budget and give a definitive explanation. However, a negative residual does not necessarily imply that sea ice is a net sink of freshwater in a warming world.

The most likely cause is that the atmospheric fields (P-E) represent fluxes over the entire grid cell surface (open ocean + sea ice). Precipitation falling on sea ice, and evaporation/sublimation from sea ice or snow, do not directly enter the total ocean flux (wfo). These fluxes change the mass of the sea-ice/snow layer, not the ocean, unless the ice melts locally. Thus, when we compute P-E using atmospheric fields, even with land masked out, we inadvertently include fluxes that may not reach the liquid ocean. This can cause P−E+R to be systematically larger than wfo in regions with seasonal or intermittent sea-ice cover. In addition, advection of sea ice out of the domain is not accounted for. Even in a warming climate with declining Antarctic sea-ice volume, the 60–90°S region can still be a net sink of freshwater if ice forms near the continent and is exported northward to melt outside our domain. This would also contribute to a negative residual without implying local sea-ice expansion. Finally, we cannot entirely rule out model-specific flux-correction terms (e.g., salinity restoring), because the corresponding diagnostics are not available. If present, such corrections could contribute to a mismatch between wfo and the individually diagnosed budget terms.

In summary, we likely overestimate P-E because the atmospheric fields include fluxes over sea ice, and it is physically plausible for sea-ice processes and ice export to produce a negative residual in this region. Ideally we would close the budget to provide a definitive answer, but this is not possible with the available data. A sentence summarizing what we explain here has been added to the manuscript. For the SOFIA simulations discussed in Pauling et al., the situation is different: in those experiments sea ice cover expands consistently in all models and is a substantial freshwater sink because strengthened stratification and surface cooling allow sea-ice cover to expand. In SSP5-8.5 this signal is superseded by the strong warming trend.

**Minor comments**

L. 114 - Remove "both" since you then have 3 references after
"Both" has been removed and commas added.

L. 172 - Missing citation
Corrected.

L. 757-759 - I don't understand the flow of this sentence. Misplaced commas?

Good catch. The sentence has been rephrased to: "Local shelf-break dynamics, including a strengthened Antarctic Slope Front (ASF) that limits Circumpolar Deep Water (CDW) intrusion, an accelerated Antarctic Slope Current (ASC) that advects anomalous Weddell waters, and reduced Dense Shelf Water (DSW) formation, play a crucial role in shaping regional responses".